# EXPLAINABLE SELF-SUPERVISED LEARNING BY SPIKING FUNCTIONS: A THEORY

## ABSTRACT

Deep neural networks trained in an end-to-end manner have been proven to be efficient in a wide range of machine learning tasks. However, there is one drawback of end-to-end learning: The learned features and information are implicitly represented in neural network parameters, which are not explainable: The learned features cannot be used as explicit regularities to explain the data probability distribution. To resolve this issue, we propose in this paper a new machine learning theory, which describes in mathematics what are 'non-randomness' and 'regularities' in a data probability distribution. Our theory applies a spiking function to distinguish data samples from random noises. In this process, 'non-randomness', or a large amount of information, is encoded by the spiking function into regularities, a small amount of information. Then, our theory describes the application of multiple spiking functions to the same data distribution. In this process, we claim that the 'best' regularities, or the optimal spiking functions, are those who can capture the largest amount of information from the data distribution, and then encode the captured information into the smallest amount of information. By optimizing the spiking functions, one can achieve an explainable self-supervised learning system.

## 1 INTRODUCTION

In the past decades, deep neural networks have being brought huge success to a wide range of machine learning tasks (LeCun et al., 1998; Vaswani et al., 2017; Devlin et al., 2018; Goodfellow et al., 2014; Rombach et al., 2022; He et al., 2016). Convolutional neural networks (CNNs) revolutionized computer vision (LeCun et al., 1998), leading to groundbreaking results in image classification, object detection, and segmentation (Deng et al., 2009; Ronneberger et al., 2015). CNNs became the backbone of many applications, from medical imaging to autonomous driving. Also, recurrent neural networks (RNNs) and their variants (Sherstinsky, 2020), such as long short-term memory (LSTM) networks and gated recurrent units (GRUs) (Hochreiter & Schmidhuber, 1997; Chung et al., 2014), combining with semantic embeddings, made significant strides in sequence modeling tasks, including language modeling, speech recognition, and time-series prediction (Zhou & Xu, 2015; Graves et al., 2013; 2006; Bengio et al., 2000; Mikolov et al., 2013).

The introduction of attention mechanisms and transformers revolutionized NLP by improving the handling of long-range dependencies and performance (Vaswani et al., 2017). Models like BERT (Devlin et al., 2018) set new benchmarks in language tasks, driven by pre-training strategies widely used in transformer models (Radford et al., 2018). Pre-training techniques, such as masked language modeling (Salazar et al., 2019) and next-word prediction (Qi et al., 2020), have propelled models like BERT, RoBERTa, ELECTRA, and T5 to excel across tasks (Devlin et al., 2018; Liu et al., 2019; Clark et al., 2020; Raffel et al., 2020). This has paved the way for large language models (LLMs), including ChatGPT (Kasneci et al., 2023; Wu et al., 2023), which excel in applications like customer service and content creation (Ray, 2023).

All these models, supervised or unsupervised, pre-trained or not, are based on an end-to-end learning process (Carion et al., 2020): The deep neural network is trained to map input data to corresponding targets, which can be labels in a supervised learning approach or masked data in an unsupervised learning approach. The loss, which is the difference between the neural network output and the target, is back propagated through the network to update the trainable parameters (Rumelhart et al., 1986).

However, there is one drawback of end-to-end learning: All the learned features and information during training are implicitly kept in network parameters, which cannot form explicit regularities to explain the data distribution. Without explicit regularities and knowledge, deep neural networks cannot emulate human-like abilities such as discovering new commonsense, generating ideas, or making plans driven by specific objectives (An et al., 2019; Sweller, 2009; Davis & Marcus, 2015).

But what are regularities in a data distribution? Or, what differs data samples from random noises? In this paper, we establish a theory to describe 'regularities' and 'non-randomness' in mathematics, which is potentially applicable to build an explainable self-supervised machine learning system.

The contribution of this paper is the proposal of such a theory, which encompasses two aspects:

1. Based on spiking functions, we define in mathematics what is non-random information, or 'non-randomness', in a data probability distribution. Then, based on information theory, we describe that regularities are a small amount of information encoding a large amount of non-random information.

2. We apply multiple spiking functions to the same data probability distribution. Then, we mathematically describe that optimal regularities are learned by spiking functions when the maximum amount of information is represented in the most concise way, or equivalently, encoded into the smallest possible amount of information. We use simple examples to show that when these optimal spiking functions are achieved, their spiking behaviors become explainable regarding the data probability distribution, and hence the system becomes an explainable self-supervised learning system.

We acknowledge that implementation models for our theory have not yet been developed. So, there are no experimental results presented in this paper. In the rest of this paper, we will first briefly introduce some related works in Section 2. Then, we describe our theory in Section 3. Finally, we conclude with a summary of this paper in Section 4.

## 2 RELATED WORK

Our work is unique since it presents a theory that defines in mathematics what are non-randomness and regularities in a data probability distribution, which can be potentially applied to practical machine learning tasks. To the edge of our knowledge, this is the first work presenting such a theory. But our theory indeed depends on spiking functions (or spiking neural networks in practice) and information theory.

Spiking Neural Networks (SNNs) have gained attention for their ability to closely mimic brain processes compared to traditional neural networks (Brown et al., 2004; y Arcas & Fairhall, 2003). Spike-timing-dependent plasticity (STDP), introduced by (Debanne & Inglebert, 2023), offers a key learning rule where synaptic strengths adjust based on spike timing. (Maass et al., 2002) then proposed the Liquid State Machine, showing how dynamic neural circuits process information. (Gütig & Sompolinsky, 2006) advanced this with the Tempotron model, where neurons learn to discriminate spatiotemporal spike patterns. Finally, (Sengupta et al., 2019) discussed converting traditional networks to SNNs, enabling energy-efficient implementations on neuromorphic hardware. Combining all these works, in our theory, given a vector $X$ as the input to a function $f$, we regard $f$ to spike on $X$ if $f(X) > 0$. This is a very simple setting comparing to many spiking neural networks.

Information theory, foundational to modern communication and data science, was pioneered by Claude E. Shannon in his paper A Mathematical Theory of Communication (Shannon, 1948). Shannon introduced key concepts like entropy, which measures uncertainty, and mutual information, which quantifies the information gained about one variable from another. Subsequent influential works include the Viterbi algorithm by (Viterbi, 1967), critical for error-correcting codes, and (Slepian & Wolf, 1973), which explored trade-offs between compression and fidelity in data transmission. (Fitts, 1954) also applied information theory to human-computer interaction. Based on the solid ground established by these pioneered works, our work applies information theory to describe non-randomness and regularities in the data distribution discovered by spiking functions.

## 3 LEARNING REGULARITIES FROM DATA USING SPIKING FUNCTIONS

We describe our theory with details in this section. First, by applying a single spiking function to the data distribution, we describes in mathematics how to measure non-randomness and what

are regularities. After that, we describe how to apply multiple spiking functions to the same data distribution. Finally, we provide a hypothesis which defines optimal spiking functions (regularities) to a data distribution. One can achieve an explainable self-supervised learning system simply by converging to the optimal spiking functions.

### 3.1 Non-randomness versus Regularities

Given a finite-dimensional vector space $\mathbf{X}$, suppose our data space is a bounded sub-region $\mathbf{S} \subset \mathbf{X}$. Suppose we have a data probability distribution $\mathbf{P}$ defined on $\mathbf{S}$. Our goal is to learn the regularities from $\mathbf{P}$. In other words, we want to know what distinguishes $\mathbf{P}$ from a random probability distribution $\mathbf{P}'$ (such as a uniform distribution) on $\mathbf{S}$. Inspired by Noise Contrastive Estimation (Gutmann & Hyvärinen, 2010), our theory focuses on distinguishing samples generated from each of these two distributions using functions.

That is, suppose we have $N$ data samples $\{X_1, X_2, \cdots, X_N\}$ generated by $\mathbf{P}$, and the same number of random samples $\{X_1', \cdots, X_N'\}$ generated by $\mathbf{P}'$. Suppose we have a function $f : \mathbf{S} \to \mathbb{R}$ that maps any vector $X \in \mathbf{S}$ to a real scalar. Inspired by the way neurons fire (or 'spike') in response to specific biochemical signals (Brown et al., 2004), our desired function $f$ should exhibit a higher response rate to data samples than to random ones. Regarding $f(X) > 0$ as a **spike** of $f$ on $X$, we aim for the spiking frequency of $f$ on $\{X_n\}_{n=1}^N$ to differ significantly from that of $f$ on $\{X_n'\}_{n=1}^N$.

We use $M$ to denote the number of observed spikes when implementing $f$ on the data samples $\{X_n\}_{n=1}^N$ generated by $\mathbf{P}$. Similarly, we use $M'$ to denote the number of spikes when implementing $f$ on random samples $\{X_n'\}_{n=1}^N$ generated by $\mathbf{P}'$. With $\widehat{p} = M/N$ and $\widehat{p'} = M'/N$, we have that $\widehat{P} = (\widehat{p}, 1 - \widehat{p})$ is the observed spiking probability distribution of $f$ on the data samples $\{X_n\}_{n=1}^N$, while $\widehat{P}' = (\widehat{p'}, 1 - \widehat{p'})$ is that of $f$ on the random samples $\{X_n'\}_{n=1}^N$. Then, we can obtain the Kullback-Leibler divergence (KL-divergence) (Hershey & Olsen, 2007) of $\widehat{P}$ over $\widehat{P}'$ as:

$$D_{KL}(\widehat{P}||\widehat{P}') = \widehat{p}\log(\frac{\widehat{p}}{\widehat{p'}}) + (1 - \widehat{p})\log(\frac{1 - \widehat{p}}{1 - \widehat{p'}}) = \frac{M}{N}\log(\frac{M}{M'}) + \frac{N - M}{N}\log(\frac{N - M}{N - M'}). \quad (1)$$

According to information theory (Shannon, 1948), $D_{KL}(\widehat{P}||\widehat{P}')$ measures the amount of information obtained if we use $\widehat{P}$ instead of $\widehat{P}'$ to estimate the spiking probability distribution of $f$ on the data samples (Shlens, 2014). Or equivalently, $f$ captures the amount of information $D_{KL}(\widehat{P}||\widehat{P}')$ from the data distribution $\mathbf{P}$ by comparing $\mathbf{P}$ with the random distribution $\mathbf{P}'$. We use $D_{KL}(\widehat{P}||\widehat{P}')$ to measure the **non-randomness** captured by $f$ from $\mathbf{P}$. That says, we define non-randomness to be the meaningful or valuable amount of information which differs a data probability distribution from a random probability distribution.

Define $p = \lim_{N \to \infty} \widehat{p} = \lim_{N \to \infty} \frac{M}{N}$, and $p' = \lim_{N \to \infty} \widehat{p'} = \lim_{N \to \infty} \frac{M'}{N}$. That is, the limits of $\widehat{p}$ and $\widehat{p'}$ over $N$ define the theoretical spiking probabilities of $f$ on data and random samples, respectively. Accordingly, we define the limit of $D_{KL}(\widehat{P}||\widehat{P}')$ over $N$ as the **theoretical spiking efficiency** of function $f$, denoted as $SE_f$. We use $D_{KL}(\widehat{P}||\widehat{P}')$ to define the **observed spiking efficiency** of function $f$, denoted as $\widehat{SE}_f$. That is,

$$SE_f = \lim_{N \to \infty} \left( \frac{M}{N}\log(\frac{M}{M'}) + \frac{N - M}{N}\log(\frac{N - M}{N - M'}) \right) \quad (2)$$

$$\widehat{SE}_f = \frac{M}{N}\log(\frac{M + \alpha}{M' + \alpha}) + \frac{N - M}{N}\log(\frac{N - M + \alpha}{N - M' + \alpha}) \quad (3)$$

According to Gibbs' inequality, we always have $D_{KL}(\widehat{P}||\widehat{P}') \geq 0$, and $D_{KL}(\widehat{P}||\widehat{P}') = 0$ if and only if $\widehat{P} = \widehat{P}'$ (Jaynes et al., 1965). However, in practice, we may occasionally have $M' = 0$ or $M' = N$, which makes $D_{KL}(\widehat{P}||\widehat{P}') = \infty$. Or, we may have $M = 0$ or $M = N$, which result in $0\log 0$. Hence, we add a small positive scalar $\alpha$ to the format of $\widehat{SE}_f$ to avoid these cases.

Both $SE_f$ and $\widehat{SE}_f$ measure (in theory and by observation, respectively) the amount of information that $f$ captures from the data distribution $\mathbf{P}$ by comparing $\mathbf{P}$ with the random distribution $\mathbf{P}'$.

Intuitively, a larger $\widehat{SE}_f$ by observation, or a larger $SE_f$ in theory indicates that $f$ can capture more information from the data distribution $\mathbf{P}$, which means that $f$ spikes more efficiently.

However, spiking efficiency itself is not enough to measure regularities learned by $f$: If $f$ itself is super complex, we may end up with over-fitting as in many deep learning experiments (Ying, 2019; Hawkins, 2004). That says, we need to define the 'size' of a function to measure its number of 'trainable' parameters, or 'conciseness':

**Definition 1.** *Suppose $a$ is a scalar parameter in the function $f$. We say that $a$ is **adjustable**, if we can adjust the value of $a$ without changing the computational complexity of $f$.*

**Definition 2.** *The **size** of a function $f$, denoted as $|f|$, is defined as the number of adjustable parameters in $f$.*

In this paper, we always calculate the size of a function using its format with the lowest computational complexity. For example, we have $|f| = 2$ for $f(x) = a \log x + b = \log x^a + b$. Also, we always assume $|f| \geq 1$ for any function $f$, even if $f$ contains no adjustable parameters.

With $|f|$ denoting the size of function $f$, we define the **conciseness** of $f$ to be $C_f = |f|^{-1}$. That says, we intuitively regard a function with a small size (or limited number of parameters) as a concise one. Then, we define the **theoretical ability** of function $f$ to be $A_f = SE_f \cdot C_f$. Similarly, we define the **observed ability** of $f$ to be $\widehat{A}_f = \widehat{SE}_f \cdot C_f$. It is easy to see that $\lim_{N \to \infty} \widehat{A}_f = A_f$. That is,

$$A_f = \lim_{N \to \infty} \left( \frac{M}{N} \log(\frac{M}{M'}) + \frac{N-M}{N} \log(\frac{N-M}{N-M'}) \right) \cdot \frac{1}{|f|} \tag{4}$$

$$\widehat{A}_f = \left( \frac{M}{N} \log(\frac{M+\alpha}{M'+\alpha}) + \frac{N-M}{N} \log(\frac{N-M+\alpha}{N-M'+\alpha}) \right) \cdot \frac{1}{|f|} \tag{5}$$

The size of $f$ defined in our theory essentially aligns with Kolmogorov complexity (Li et al., 2008), which describes the minimum amount of information required to specify $f$ unambiguously (Wallace & Dowe, 1999). So, intuitively, the ability (theoretical or observed) of function $f$ measures the amount of information captured by $f$ relative to the amount of information required to specify $f$. A function $f$ with high ability implies that $f$ is able to encode a large amount of information into a small amount of information, which is why we use 'ability' to denote this variable.

Summarizing our discussion in this section, we define that ***regularities** are a small amount of information representing a large amount of information*. A spiking function with higher ability learns stronger or better regularities. Also, we do not strictly differ a spiking function from the regularities learned by that function. In the rest of this paper, we in most cases regard function $f$ as its learned regularities, and vice versa.

In the following sections, we describe how to expand our theory to multiple spiking functions. But we will first introduce some mathematical basis in the next sub-section.

## 3.2 BASIC CONCEPTS

We assume by default in this paper that the vector space $\mathbf{X}$ is either real or complex (i.e., $\mathbf{X} = \mathbb{R}^m$ or $\mathbf{X} = \mathbb{C}^m$ with some finite integer $m$). Then, we adopt the Euclidean distance as the metric on $\mathbf{X}$ (Gower, 1985), which defines the distance $d(X, Y)$ between any two vectors $X, Y \in \mathbf{X}$. A little bit more detailed discussion can be found in Appendix A.

Then, we use the Lebesgue measure (Bartle, 2014) to define the 'volume' of a subset $\mathbf{E} \subset \mathbf{X}$. The Lebesgue measure is a regular choice to measure volumes of subsets within a finite-dimensional real or complex vector space (Ciesielski, 1989), which coincides with our usual understanding of volume. Say, when $\mathbf{X} = \mathbb{R}^3$, the Lebesgue measure of a sphere with radius $r$ is $\frac{3}{4}\pi r^3$. A more detailed definition on Lebesgue measure is provided in Appendix A.

With these concepts, we can then evaluate the spiking efficiency, conciseness and ability of a function in a straightforward way. That is, we will first have:

**Definition 3.** *Suppose $\mathbf{X}$ is a finite-dimensional real or complex vector space, and suppose $\mathbf{S} \subset \mathbf{X}$ is a bounded sub-region in $\mathbf{X}$. Suppose $f : \mathbf{S} \to \mathbb{R}$ is a function defined on $\mathbf{S}$. We define the **spiking region** of $f$, denoted by $\mathbf{S}_f$, to be the sub-region in $\mathbf{S}$ consisting of the vectors that $f$ spikes on. That is, $\mathbf{S}_f = \{X \in \mathbf{S} | f(X) > 0\}$.*

*Also, we say that two spiking regions are **distinct**, if there exists a vector $X \in \mathbf{S}$ that belongs to one spiking region but does not belong to the other.*

Obviously, there is an unique spiking region $\mathbf{S}_f$ to any function $f$. We will work with many types of spiking regions in the rest of this paper. We note that spiking region is only an analytical tool we created in the theory. In practice, it is very difficult and not meaningful to accurately measure the spiking region of a function, especially in high dimensional data spaces.

When function $f$ is continuous regarding the metric on the vector space $\mathbf{X}$, we have the following lemma to guarantee that $\mathbf{S}_f$ is Lebesgue-measurable (i.e., the region indeed has a determined volume under Lebesgue measure):

**Lemma 1.** *Suppose $\mathbf{X}$ is a finite-dimensional real or complex vector space, and suppose $\mathbf{S} \subset \mathbf{X}$ is a bounded sub-region in $\mathbf{X}$. Suppose $f : \mathbf{S} \to \mathbb{R}$ is a continuous function defined on $\mathbf{S}$ (i.e., for any $\epsilon > 0$, there exists a $\delta > 0$ such that for any $X, Y \in \mathbf{S}$, $d(X, Y) < \delta$ implies $|f(X) - f(Y)| < \epsilon$). Then, the spiking region $\mathbf{S}_f = \{X \in \mathbf{S}| f(X) > 0\}$ is always Lebesgue-measurable.*

The discussion on Lebesgue-measurable/non-measurable sets as well as the proof of this lemma are provided in Appendix A. Lebesgue non-measurable sets indeed exist in a finite-dimensional vector space. However, a non-measurable set is usually extremely complicated. The Vitali set (Kharazishvili, 2011) is a famous non-measurable set in $\mathbb{R}$, which is discussed in Appendix A. To be specific, we believe that any function $f$ with a finite size cannot possess a non-measurable spiking region. That is,

**Hypothesis 1.** *Suppose $\mathbf{X}$ is a finite-dimensional real or complex vector space, and suppose $\mathbf{S} \subset \mathbf{X}$ is a bounded sub-region in $\mathbf{X}$. Suppose $f : \mathbf{S} \to \mathbb{R}$ is a function defined on $\mathbf{S}$ with a finite size (i.e., there are finite adjustable parameters in $f$). Then, the spiking region $\mathbf{S}_f = \{X \in \mathbf{S}| f(X) > 0\}$ is always Lebesgue-measurable.*

Based on this hypothesis, we claim that: Any function with a finite size can only capture finite amount of information from a data distribution, as long as the data distribution is a 'regular' one. That is:

**Theorem 1.** *Suppose $\mathbf{X}$ is a finite-dimensional real or complex vector space, and suppose $\mathbf{S} \subset \mathbf{X}$ is a bounded sub-region in $\mathbf{X}$. Suppose we have the data probability distribution $\mathbf{P}$ defined on $\mathbf{S}$, with the probability density function to be $g(X)$. Furthermore, suppose there exists an upper bound $\Omega < \infty$, such that $g(X) \leq \Omega$ for any $X \in \mathbf{S}$. Finally, suppose we have the random distribution $\mathbf{P}'$ to be the uniform distribution defined on $\mathbf{S}$.*

*Then, for any function $f : \mathbf{S} \to \mathbb{R}$ with a finite size $|f|$, its theoretical spiking efficiency $SE_f$ obtained with respect to $\mathbf{P}$ and $\mathbf{P}'$ is bounded by $0 \leq SE_f \leq \Omega \cdot |\mathbf{S}| \cdot \log(\Omega \cdot |\mathbf{S}|)$, where $|\mathbf{S}|$ is the Lebesgue measure of data space $\mathbf{S}$.*

Again, we provide the detailed proof of this theorem in Appendix A. However, we want to provide the following formulas regarding $SE_f$ here (in which $\mathbf{S}_f^c = \mathbf{S} \backslash \mathbf{S}_f$ is the complement of the spiking region $\mathbf{S}_f$ in $\mathbf{S}$, $g'(X) \equiv \frac{1}{|\mathbf{S}|}$ is the probability density function of the uniform distribution $\mathbf{P}'$, and $|\mathbf{S}_f|$ is the Lebesgue measure of $\mathbf{S}_f$):

$$SE_f = \left( \int_{\mathbf{S}_f} g(X) \, dX \right) \cdot \log \left( \frac{\int_{\mathbf{S}_f} g(X) \, dX}{\int_{\mathbf{S}_f} g'(X) \, dX} \right) + \left( \int_{\mathbf{S}_f^c} g(X) \, dX \right) \cdot \log \left( \frac{\int_{\mathbf{S}_f^c} g(X) \, dX}{\int_{\mathbf{S}_f^c} g'(X) \, dX} \right) \quad (6)$$

$$= \left( \int_{\mathbf{S}_f} g(X) \, dX \right) \cdot \log \left( \frac{|\mathbf{S}| \int_{\mathbf{S}_f} g(X) \, dX}{|\mathbf{S}_f|} \right) + \left( 1 - \int_{\mathbf{S}_f} g(X) \, dX \right) \cdot \log \left( \frac{|\mathbf{S}| - |\mathbf{S}| \int_{\mathbf{S}_f} g(X) \, dX}{|\mathbf{S}| - |\mathbf{S}_f|} \right) \quad (7)$$

By a 'regular' data distribution $\mathbf{P}$, we mean that there exists an upper bound $\Omega < \infty$ for the probability density function $g(X)$ of $\mathbf{P}$. Otherwise, $\mathbf{P}$ will contain singularities with infinitely large probability density (Meunier & Villermaux, 2007). In such a case our theorem is not true. In the rest part, we always use the uniform distribution defined on the entire data space $\mathbf{S}$ as the random distribution $\mathbf{P}'$.

### 3.3 APPLYING MULTIPLE SPIKING FUNCTIONS TO THE DATA DISTRIBUTION

In this sub-section, we discuss the situation of multiple spiking functions being applied to the same data distribution.

Again, suppose we have the data distribution $\mathbf{P}$ and random (uniform) distribution $\mathbf{P}'$ defined on the data space $\mathbf{S}$. Suppose we have a sequence of functions $\mathbf{f} = (f_1, \cdots, f_K)$, where each function $f_k : \mathbf{S} \to \mathbb{R}$ has a finite size $|f_k|$. Given the data samples $\{X_n\}_{n=1}^N$ generated by $\mathbf{P}$ and random samples $\{X_n'\}_{n=1}^N$ generated by $\mathbf{P}'$, suppose function $f_1$ spikes on $M_1$ data samples and $M_1'$ random samples, respectively. Remember that the objective of each function is to capture information from the data distribution by discovering non-randomness. Hence, if $f_1$ already spikes on a vector $X$ (either a data sample or a random sample), it will be meaningless to consider whether $f_2$ spikes on $X$. This is because the corresponding non-randomness, or non-random features related to $X$ have already been discovered by $f_1$, so that it becomes meaningless for $f_2$ to discover these non-random features again.

As a result, after ignoring the data samples that $f_1$ spikes on, suppose there are $M_2$ data samples in $\{X_n\}_{n=1}^N$ making $f_2$ spike. Similarly, suppose there are $M_2'$ random samples in $\{X_n'\}_{n=1}^N$ making $f_2$ spike while not making $f_1$ spike. Carrying on this process, suppose there are $M_k$ data samples in $\{X_n\}_{n=1}^N$ and $M_k'$ random samples in $\{X_n'\}_{n=1}^N$ respectively that make $f_k$ spike, but do not make $f_1, \cdots, f_{k-1}$ spike. In this way, we can define the **theoretical spiking efficiency** $SE_{f_k}$ and **observed spiking efficiency** $\widehat{SE}_{f_k}$ for each function $f_k$ in $\mathbf{f} = (f_1, \cdots, f_K)$ as:

$$SE_{f_k} = \lim_{N \to \infty} \left( \frac{M_k}{N} \log(\frac{M_k}{M_k'}) + \frac{N - M_k}{N} \log(\frac{N - M_k}{N - M_k'}) \right) \tag{8}$$

$$\widehat{SE}_{f_k} = \frac{M_k}{N} \log(\frac{M_k + \alpha}{M_k' + \alpha}) + \frac{N - M_k}{N} \log(\frac{N - M_k + \alpha}{N - M_k' + \alpha}). \tag{9}$$

Similar to formula 2, $\alpha$ is a small positive number to avoid $\log(0)$ or $M/0$ in practice.

Then, suppose $M_{\mathbf{f}} = \sum_{k=1}^K M_k$ and $M_{\mathbf{f}}' = \sum_{k=1}^K M_k'$. Since there is no overlapping on spiked samples when calculating $M_k$ and $M_k'$, we have that $M_{\mathbf{f}}$ is the total number of data samples in $\{X_n\}_{n=1}^N$ that make at least one function in $\mathbf{f}$ to spike. Similarly, $M_{\mathbf{f}}'$ is the total number of random samples in $\{X_n'\}_{n=1}^N$ making at least one function in $\mathbf{f}$ to spike. We define the **theoretical spiking efficiency** and **observed spiking efficiency** of $\mathbf{f} = (f_1, \cdots, f_K)$ as:

$$SE_{\mathbf{f}} = \lim_{N \to \infty} \left( \frac{M_{\mathbf{f}}}{N} \log(\frac{M_{\mathbf{f}}}{M_{\mathbf{f}}'}) + \frac{N - M_{\mathbf{f}}}{N} \log(\frac{N - M_{\mathbf{f}}}{N - M_{\mathbf{f}}'}) \right) \tag{10}$$

$$\widehat{SE}_{\mathbf{f}} = \frac{M_{\mathbf{f}}}{N} \log(\frac{M_{\mathbf{f}} + \alpha}{M_{\mathbf{f}}' + \alpha}) + \frac{N - M_{\mathbf{f}}}{N} \log(\frac{N - M_{\mathbf{f}} + \alpha}{N - M_{\mathbf{f}}' + \alpha}) \tag{11}$$

Intuitively, $SE_{\mathbf{f}}$ measures (in theory) the amount of information captured by the entire sequence of functions $\mathbf{f} = (f_1, \cdots, f_K)$, while $SE_{f_k}$ measures (in theory) the amount of valid information captured by each function $f_k$ in the sequence. By 'valid information', we mean the information related to the newly discovered non-randomness by $f_k$ that is not discovered by $f_1, \cdots, f_{k-1}$.

We define the **spiking region** of each $f_k$ in $\mathbf{f} = (f_1, \cdots, f_K)$ by removing the spiking regions of functions ahead of $f_k$ in the sequence. That is, $\mathbf{S}_{f_k} = \{X \in \mathbf{S} | f_k(X) > 0 \text{ and } f_i(X) \leq 0 \text{ for } i = 1, \cdots, k-1\}$. In fact, if $\mathbf{S}_{f_k}^{ind}$ denotes the spiking region of $f_k$ when it is considered as an independent function, then we have that $\mathbf{S}_{f_k} = \mathbf{S}_{f_k}^{ind} \backslash (\cup_{i=1}^{k-1} \mathbf{S}_{f_i}^{ind})$. We denote $\mathbf{S}_{f_k}^{ind}$ as the **independent spiking region** of each function $f_k$. Then, we define

$$\mathbf{S}_{\mathbf{f}} = \{X \in \mathbf{S} | f_k(X) > 0 \text{ for any function } f_k \in \{f_1, \cdots, f_K\}\}$$

to be the **spiking region** of $\mathbf{f}$. That is, $\mathbf{S}_{\mathbf{f}}$ consists of vectors that make at least one function in $\mathbf{f} = (f_1, \cdots, f_K)$ to spike. We have that $\mathbf{S}_{f_k} \cap \mathbf{S}_{f_i} = \emptyset$ if $k \neq i$, and $\mathbf{S}_{\mathbf{f}} = \cup_{k=1}^K \mathbf{S}_{f_k} = \cup_{k=1}^K \mathbf{S}_{f_k}^{ind}$. A straightforward demonstration is shown in Figure 1.

By Hypothesis 1, each $\mathbf{S}_{f_k}^{ind}$ is measurable since $|f_k|$ is finite. This means that $\mathbf{S}_{f_k} = \mathbf{S}_{f_k}^{ind} \backslash (\cup_{i=1}^{k-1} \mathbf{S}_{f_i}^{ind})$ is measurable as well. Hence, $\mathbf{S}_{\mathbf{f}} = \cup_{k=1}^K \mathbf{S}_{f_k}$ is also measurable. Following the same proof method as Theorem 1, we have:

**Theorem 2.** *Suppose $\mathbf{X}$ is a finite-dimensional real or complex vector space, and suppose $\mathbf{S} \subset \mathbf{X}$ is a bounded sub-region in $\mathbf{X}$. Suppose there are the data probability distribution $\mathbf{P}$ and the uniform distribution $\mathbf{P}'$ defined on $\mathbf{S}$. Also, suppose the probability density function $g(X)$ of $\mathbf{P}$ is bounded by a finite number $\Omega$ (i.e., $\mathbf{P}$ is regular).*

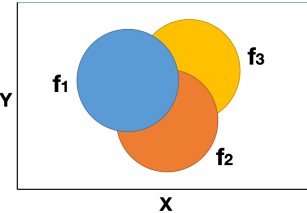

Figure 1: The spiking regions of functions in $\mathbf{f} = (f_1, f_2, f_3)$ defined on the xy-plane. $\mathbf{S}_{f_1}$ is the blue circle in the front. $\mathbf{S}_{f_2}$ is the red circle but removing overlapping with $\mathbf{S}_{f_1}$. Then, $\mathbf{S}_{f_3}$ is the yellow circle after removing $\mathbf{S}_{f_1}$ and $\mathbf{S}_{f_2}$.

*Suppose $\mathbf{f} = (f_1, \cdots, f_K)$ is a sequence of functions with each function $f_k : \mathbf{S} \to \mathbb{R}$ possessing a finite size $|f_k|$. Then, with respect to $\mathbf{P}$ and $\mathbf{P}'$, the theoretical spiking efficiencies of both $\mathbf{f}$ and each $f_k$ are bounded. That is, we have $0 \le SE_{\mathbf{f}} \le \Omega \cdot |\mathbf{S}| \cdot \log(\Omega \cdot |\mathbf{S}|)$, as well as $0 \le SE_{f_k} \le \Omega \cdot |\mathbf{S}| \cdot \log(\Omega \cdot |\mathbf{S}|)$ for $k = 1, \cdots, K$. Here, $|\mathbf{S}|$ is the Lebesgue measure of data space $\mathbf{S}$.*

We note that independent with the order of functions in $\mathbf{f} = (f_1, \cdots, f_K)$, once the set of functions $\{f_1, \cdots, f_K\}$ is fixed, the theoretical spiking efficiency $SE_{\mathbf{f}}$ and the spiking region $\mathbf{S}_{\mathbf{f}}$ will be determined. In fact, suppose $\mathbf{S}_{\mathbf{f}}^c = \mathbf{S} \backslash \mathbf{S}_{\mathbf{f}}$ denotes the complement of $\mathbf{S}_{\mathbf{f}}$ in $\mathbf{S}$, $g'(X) \equiv \frac{1}{|\mathbf{S}|}$ denotes the probability density function of $\mathbf{P}'$, and $|\mathbf{S}_{\mathbf{f}}|$ denotes the Lebesgue measure of $\mathbf{S}_{\mathbf{f}}$. Then, it can be derived from the same proof method of Theorem 1 that:

$$SE_{\mathbf{f}} = \left( \int_{\mathbf{S}_{\mathbf{f}}} g(X)\, dX \right) \cdot \log\left( \frac{\int_{\mathbf{S}_{\mathbf{f}}} g(X)\, dX}{\int_{\mathbf{S}_{\mathbf{f}}} g'(X)\, dX} \right) + \left( \int_{\mathbf{S}_{\mathbf{f}}^c} g(X)\, dX \right) \cdot \log\left( \frac{\int_{\mathbf{S}_{\mathbf{f}}^c} g(X)\, dX}{\int_{\mathbf{S}_{\mathbf{f}}^c} g'(X)\, dX} \right) \tag{12}$$

$$= \left( \int_{\mathbf{S}_{\mathbf{f}}} g(X)\, dX \right) \cdot \log\left( \frac{|\mathbf{S}| \int_{\mathbf{S}_{\mathbf{f}}} g(X)\, dX}{|\mathbf{S}_{\mathbf{f}}|} \right) + \left( 1 - \int_{\mathbf{S}_{\mathbf{f}}} g(X)\, dX \right) \cdot \log\left( \frac{|\mathbf{S}| - |\mathbf{S}| \int_{\mathbf{S}_{\mathbf{f}}} g(X)\, dX}{|\mathbf{S}| - |\mathbf{S}_{\mathbf{f}}|} \right) \tag{13}$$

Also, replacing $\mathbf{f}$ by $f_k$ in these formulas, we can get the formulas of $SE_{f_k}$ for each $f_k$ in $\mathbf{f} = (f_1, \cdots, f_K)$.

The above formulas actually imply that: If $\mathbf{f} = (f_1, \cdots, f_K)$ and $\widetilde{\mathbf{f}} = (\widetilde{f}_1, \cdots, \widetilde{f}_{\widetilde{K}})$ are two sequences of finite-sized functions defined on $\mathbf{S}$, we will then have $SE_{\mathbf{f}} = SE_{\widetilde{\mathbf{f}}}$ as long as their spiking regions $\mathbf{S}_{\mathbf{f}}$ and $\mathbf{S}_{\widetilde{\mathbf{f}}}$ are coincide. But the inverse is not true: If two sequences of finite-sized functions have the same theoretical spiking efficiency, their spiking regions may still be distinct. The following definition provides a more throughout description:

**Definition 4.** *Suppose $\mathbf{X}$ is a finite-dimensional real or complex vector space, and suppose $\mathbf{S} \subset \mathbf{X}$ is a bounded sub-region in $\mathbf{X}$. Suppose there are the data probability distribution $\mathbf{P}$ and the uniform distribution $\mathbf{P}'$ defined on $\mathbf{S}$. Also, suppose the probability density function $g(X)$ of $\mathbf{P}$ is bounded by a finite number $\Omega$ (i.e., $\mathbf{P}$ is regular).*

*Suppose $\mathbf{f} = (f_1, \cdots, f_K)$ and $\widetilde{\mathbf{f}} = (\widetilde{f}_1, \cdots, \widetilde{f}_{\widetilde{K}})$ are two sequences of finite-sized functions defined on $\mathbf{S}$. We say that $\mathbf{f}$ and $\widetilde{\mathbf{f}}$ are **spiking equivalent** with respect to $\mathbf{P}$ and $\mathbf{P}'$, denoted as $\mathbf{f} \sim \widetilde{\mathbf{f}}$, if $SE_{\mathbf{f}} = SE_{\widetilde{\mathbf{f}}}$.*

*Suppose $\mathbf{f} = (f_1, \cdots, f_K)$ is a sequence of finite-sized functions defined on $\mathbf{S}$. We define the **spiking equivalence class** of $\mathbf{f}$, denoted as $\mathcal{E}_{\mathbf{f}}$, to be the set consisting of all the sequences of finite-sized functions that are spiking equivalent to $\mathbf{f}$. That is,*

$$\mathcal{E}_{\mathbf{f}} = \{ \widetilde{\mathbf{f}} = (\widetilde{f}_1, \cdots, \widetilde{f}_{\widetilde{K}}) \big| |\widetilde{f}_k| < \infty \text{ for } k = 1, \cdots, \widetilde{K}; \text{ and } SE_{\mathbf{f}} = SE_{\widetilde{\mathbf{f}}} \},$$

*where $K$ and $\widetilde{K}$ are not necessarily equal, and different values of $\widetilde{K}$ are allowed in $\mathcal{E}_{\mathbf{f}}$.*

*Finally, if there exists a sequence of finite-sized functions $\mathbf{f}^* = (f_1^*, \cdots, f_{K^*}^*)$, such that for any sequence of finite-sized functions $\widetilde{\mathbf{f}} = (\widetilde{f}_1, \cdots, \widetilde{f}_{\widetilde{K}})$, the inequality $SE_{\mathbf{f}^*} \ge SE_{\widetilde{\mathbf{f}}}$ always holds true, then we call $\mathbf{f}^*$ the **most efficient encoder** of $\mathbf{P}$ based on $\mathbf{P}'$, denoted as $\mathbf{f}_{\mathbf{P}, \mathbf{P}'}^*$. We call $\mathcal{E}_{\mathbf{f}_{\mathbf{P}, \mathbf{P}'}^*}$, the spiking equivalence class of $\mathbf{f}_{\mathbf{P}, \mathbf{P}'}^*$, the **most efficient class** of $\mathbf{P}$ based on $\mathbf{P}'$, denoted as $\mathcal{E}_{\mathbf{P}, \mathbf{P}'}^*$.*

We note that there is no guarantee on the existence of a most efficient encoder. A regular data distribution that does not possess a most efficient encoder (i.e., an empty most efficient class) is discussed at the end of Appendix B. Also, we believe that:

**1:** Any existing spiking equivalence class contains infinite elements (sequences of finite-sized functions), which is formally described by **Hypothesis 3** in Appendix A.

**2:** Upon the existence of the most efficient class, every most efficient encoder shall have exactly the same spiking region. We formally describe this by **Hypothesis 4** in Appendix A.

### 3.4 OPTIMAL ENCODER OF THE DATA DISTRIBUTION

Based on the previous discussion, we can now consider the ability of multiple functions.

Again, suppose we have the regular data distribution $\mathbf{P}$ and uniform distribution $\mathbf{P}'$ defined on the data space $\mathbf{S}$. Suppose $\mathbf{f} = (f_1, \cdots, f_K)$ is a sequence of functions defined on $\mathbf{S}$, where each function $f_k : \mathbf{S} \to \mathbb{R}$ has a finite size $|f_k|$. Given $N$ data samples $\{X_n\}_{n=1}^N$ generated by $\mathbf{P}$ and $N$ random samples $\{X_n'\}_{n=1}^N$ generated by $\mathbf{P}'$, suppose there are $M_k$ data samples and $M_k'$ random samples respectively that make $f_k$ spike, but do not make $f_1, \cdots, f_{k-1}$ spike. Using $M_k$, $M_k'$ and $N$, we can obtain the theoretical spiking efficiency $SE_{f_k}$ and observed spiking efficiency $\widehat{SE}_{f_k}$ for each function $f_k$ in $\mathbf{f}$ by formula 8.

Then, we can obtain the **theoretical ability** $A_{f_k} = SE_{f_k} \cdot C_{f_k}$ and the **observed ability** $\widehat{A}_{f_k} = \widehat{SE}_{f_k} \cdot C_{f_k}$ for each $f_k$ in $\mathbf{f}$, where $C_{f_k} = |f_k|^{-1}$ is the **conciseness** of $f_k$:

$$A_{f_k} = \lim_{N \to \infty} \left( \frac{M_k}{N} \log(\frac{M_k}{M_k'}) + \frac{N - M_k}{N} \log(\frac{N - M_k}{N - M_k'}) \right) \cdot \frac{1}{|f_k|};$$

$$\widehat{A}_{f_k} = \left( \frac{M_k}{N} \log(\frac{M_k + \alpha}{M_k' + \alpha}) + \frac{N - M_k}{N} \log(\frac{N - M_k + \alpha}{N - M_k' + \alpha}) \right) \cdot \frac{1}{|f_k|}. \tag{14}$$

In our opinion, the ability (theoretical or observed) indicates the 'effort' or 'work' function $f$ made to encode the captured information into its parameters. As a result, the ability, or 'work', of the entire sequence of functions $\mathbf{f} = (f_1, \cdots, f_K)$ should be the sum of that from each function.

That says, with respect to $\mathbf{P}$ and $\mathbf{P}'$, we define the **theoretical ability** of the sequence of functions $\mathbf{f} = (f_1, \cdots, f_K)$ to be $A_{\mathbf{f}} = \sum_{k=1}^K A_{f_k}$. We define the **observed ability** of $\mathbf{f} = (f_1, \cdots, f_K)$ to be $\widehat{A}_{\mathbf{f}} = \sum_{k=1}^K \widehat{A}_{f_k}$:

$$A_{\mathbf{f}} = \sum_{k=1}^K \left[ \lim_{N \to \infty} \left( \frac{M_k}{N} \log(\frac{M_k}{M_k'}) + \frac{N - M_k}{N} \log(\frac{N - M_k}{N - M_k'}) \right) \cdot \frac{1}{|f_k|} \right];$$

$$\widehat{A}_{\mathbf{f}} = \sum_{k=1}^K \left[ \left( \frac{M_k}{N} \log(\frac{M_k + \alpha}{M_k' + \alpha}) + \frac{N - M_k}{N} \log(\frac{N - M_k + \alpha}{N - M_k' + \alpha}) \right) \cdot \frac{1}{|f_k|} \right]. \tag{15}$$

According to Theorem 2, for each finite-sized function $f_k$ in $\mathbf{f}$, its theoretical spiking efficiency $SE_{f_k}$ is bounded. As discussed in Section 3.1, we require $|f| \geq 1$ for any function. Hence, we have the conciseness $C_{f_k} = |f_k|^{-1} \leq 1$ for each $f_k$ in $\mathbf{f}$. This means that the theoretical ability $A_{f_k} = SE_{f_k} \cdot C_{f_k}$ of each function $f_k$ is also bounded. Hence, the theoretical ability $A_{\mathbf{f}} = \sum_{k=1}^K A_{f_k}$ of $\mathbf{f} = (f_1, \cdots, f_K)$ is bounded.

Now, we can present a major hypothesis in our theory, which defines the optimal regularities with respect to a data probability distribution. This hypothesis is also the key to achieve an explainable self-supervised learning system.

**Hypothesis 2.** *Suppose $\mathbf{X}$ is a finite-dimensional real or complex vector space, and suppose $\mathbf{S} \subset \mathbf{X}$ is a bounded sub-region in $\mathbf{X}$. Suppose there are the data probability distribution $\mathbf{P}$ and the uniform distribution $\mathbf{P}'$ defined on $\mathbf{S}$. Also, suppose the probability density function $g(X)$ of $\mathbf{P}$ is bounded by a finite number $\Omega$ (i.e., $\mathbf{P}$ is regular).*

*Given a sequence of finite-sized functions $\mathbf{f} = (f_1, \cdots, f_K)$ defined on $\mathbf{S}$, suppose that with respect to $\mathbf{P}$ and $\mathbf{P}'$, $\mathcal{E}_{\mathbf{f}}$ is the spiking equivalence class of $\mathbf{f}$, and $\Gamma = SE_{\mathbf{f}}$ is the theoretical spiking efficiency*

*of* **f**. *Then, there exists at least one sequence of finite-sized functions* $\mathbf{f}^\dagger = (f_1^\dagger, \cdots, f_{K^\dagger}^\dagger) \in \mathcal{E}_{\mathbf{f}}$, *such that for any other* $\widetilde{\mathbf{f}} = (\widetilde{f}_1, \cdots, \widetilde{f}_{\widetilde{K}}) \in \mathcal{E}_{\mathbf{f}}$, *the inequality* $A_{\mathbf{f}^\dagger} \geq A_{\widetilde{\mathbf{f}}}$ *always holds true. We call such an* $\mathbf{f}^\dagger = (f_1^\dagger, \cdots, f_{K^\dagger}^\dagger)$ *a* $\Gamma$-*level optimal encoder of* **P** *based on* **P**′, *denoted as* $\mathbf{f}_{\mathbf{P},\mathbf{P}'}^{\dagger \sim \Gamma}$.

*Finally, suppose the most efficient class* $\mathcal{E}_{\mathbf{P},\mathbf{P}'}^*$ *with respect to* **P** *and* **P**′ *is not empty. Then, there exists at least one most efficient encoder* $\mathbf{f}^\dagger = (f_1^\dagger, \cdots, f_{K^\dagger}^\dagger) \in \mathcal{E}_{\mathbf{P},\mathbf{P}'}^*$, *such that for any other most efficient encoder* $\mathbf{f}^* = (f_1^*, \cdots, f_{K^*}^*) \in \mathcal{E}_{\mathbf{P},\mathbf{P}'}^*$, *the inequality* $A_{\mathbf{f}^\dagger} \geq A_{\mathbf{f}^*}$ *always holds true. We call such an* $\mathbf{f}^\dagger = (f_1^\dagger, \cdots, f_{K^\dagger}^\dagger)$ *an* **optimal encoder** *of* **P** *based on* **P**′, *denoted as* $\mathbf{f}_{\mathbf{P},\mathbf{P}'}^\dagger$.

Based on the above hypothesis, suppose the data distribution **P** is also uniformly distributed within specific regions of **S**. Then, given the existence of an optimal encoder $\mathbf{f}_{\mathbf{P},\mathbf{P}'}^\dagger = (f_1^\dagger, \cdots, f_{K^\dagger}^\dagger)$ with respect to **P** and **P**′, we claim that intuitively, the independent spiking regions $\{\mathbf{S}_{f_1^\dagger}^{ind}, \cdots, \mathbf{S}_{f_{K^\dagger}^\dagger}^{ind}\}$ of the functions $\{f_1^\dagger, \cdots, f_{K^\dagger}^\dagger\}$ in $\mathbf{f}_{\mathbf{P},\mathbf{P}'}^\dagger$ divide the data space **S** in the most appropriate way with respect to the data distribution **P**.

We provide several graphs in Figure 2 for a better understanding on this statement. Note that the example distribution in each graph of Figure 2 has the data space $\mathbf{S} \subset \mathbb{R}^2$. But we claim that the same statement can be applied to data probability distributions in higher dimensional vector spaces.

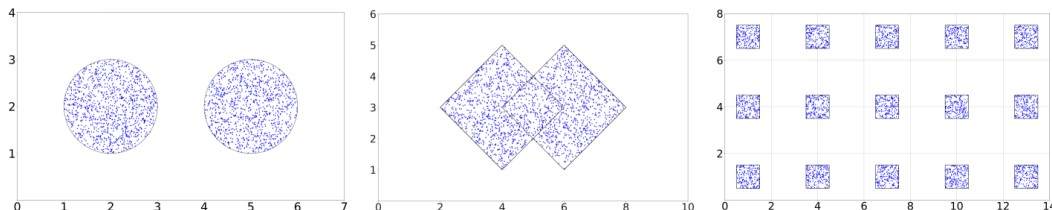

Figure 2: Optimal encoders to several simple data distributions.

In the left graph of Figure 2, we have the data distribution **P** itself to be a uniform distribution within two disjoint circles: $\sqrt{(x-2)^2 + (y-2)^2} = 1$ and $\sqrt{(x-5)^2 + (y-2)^2} = 1$. The data space **S** is the rectangle $\{x, y \mid 0 \leq x \leq 7, 0 \leq y \leq 4\} \subset \mathbb{R}^2$, and the random distribution **P**′ is uniform within **S**. In Appendix E, our evaluation will show that an optimal encoder with respect to **P** and **P**′ in this example will likely consist of two binary functions corresponding to the two circles:

$$f_1^\dagger(x,y) = \begin{cases} 1, & \text{if } \sqrt{(x-2)^2 + (y-2)^2} \leq 1; \\ 0, & \text{otherwise.} \end{cases} \quad , \quad f_2^\dagger(x,y) = \begin{cases} 1, & \text{if } \sqrt{(x-5)^2 + (y-2)^2} \leq 1; \\ 0, & \text{otherwise.} \end{cases}$$

However, we acknowledge that our evaluation is a numerical enumeration rather than a strict mathematical proof.

The middle graph of Figure 2 shows a data distribution **P** that is uniform within the area covered by two overlapped diamonds (a diamond is a $45°$ rotation from a square). The vertex of each diamond coincides with the center of the other diamond. The centers of the two diamonds are $x = 4, y = 3$ and $x = 6, y = 3$. Again, the random distribution is uniform within $\mathbf{S} = \{x, y \mid 0 \leq x \leq 10, 0 \leq y \leq 6\}$. We will show in Appendix E that an optimal encoder in this example will likely provide the independent spiking regions exactly matching with these two diamonds.

Finally, in the right graph of Figure 2, there are 15 squares within $\mathbf{S} = \{x, y \mid 0 \leq x \leq 14, 0 \leq y \leq 8\}$. The data distribution **P** is uniform within these 15 squares, while the random distribution **P**′ is uniform in **S**. Similarly, we show in Appendix E that an optimal encoder with respect to **P** and **P**′ will likely consist of 15 functions, providing 15 spiking regions fitting each of these squares. From all the three examples, we can see that $K^\dagger$, the number of functions in an optimal encoder, is naturally determined by the data distribution **P**. More example distributions and discussions can be found in Appendix E.

This property of an optimal encoder (dividing the data space in the most appropriate way regarding the data probability distribution) is actually **self-supervised explainability**: Without annotations

or labeling, one can get an explainable sequence of spiking functions $\mathbf{f} = (f_1, \cdots, f_K)$ regarding the data distribution $\mathbf{P}$ simply by maximizing the ability of $\mathbf{f}$. In this way, $\mathbf{f}$ will approach to an optimal encoder of $\mathbf{P}$ and divide the data space in the most appropriate way, or an 'explainable way', regarding $\mathbf{P}$. That is, one can get explainable spiking functions in a self-supervised manner, which can be an alternative to end-to-end learning.

In practice, the data distribution $\mathbf{P}$ may not be uniformly distributed within its specific regions. In this case, we admit that an optimal encoder obtained according to our theory will not be perfect. One should refer to the last example in Appendix E for more details. But in fact, this shows a defect of our theory: The data probability density variations within a function's spiking region cannot be appropriately represented. In other words, beyond spiking or not spiking, we need to further consider the spiking magnitude, or spiking strength, of a function on an input sample. A refined theory taking spiking strength into consideration is briefly described in Appendix D.

As we mentioned, there is no actual implementation or realization of our theory. However, we indeed design an implementation pipeline for spiking functions to converge to an optimal encoder in practice. Our designed pipeline is based on multiple bi-output functions, which is described in Appendix C.

Once again, given the sequence of functions $\mathbf{f} = (f_1, \cdots, f_K)$, its theoretical spiking efficiency $SE_{\mathbf{f}}$ measures the total amount of information captured from the data distribution by all the functions in $\mathbf{f}$. The theoretical ability $A_{f_k}$ of each function $f_k$ in $\mathbf{f}$ measures the valid effort made by $f_k$ on information encoding: The amount of valid information $SE_{f_k}$ (i.e., the amount of information related to the non-randomness that is discovered by $f_k$ but not by $f_1, \cdots, f_{k-1}$) is encoded into $|f_k|$, whereas the effort made in this encoding process is measured by $A_{f_k} = SE_{f_k}/|f_k|$. Then, the theoretical ability $A_{\mathbf{f}} = \sum_{k=1}^{K} A_{f_k}$ measures the total valid effort made by all the functions in $\mathbf{f} = (f_1, \cdots, f_K)$ on information encoding.

Therefore, given the data distribution $\mathbf{P}$ and the uniform distribution $\mathbf{P}'$, if an optimal encoder $\mathbf{f}_{\mathbf{P},\mathbf{P}'}^{\dagger}$ does exist, it is the sequence of functions that captures the largest amount of information from $\mathbf{P}$, and then encodes (or compresses) the information with the greatest effort. That says, *the optimal regularities capture the largest amount of information and represent it in the most concise way, or equivalently, encode it by the smallest amount of information.* Finally, according to these discussions, we can see that a learning system can obtain explainable and meaningful representations of a data probability distribution in a self-supervised manner, simply by encoding large amount of information into small amount of information.

## 4 CONCLUSION

In this paper, we establish a theory on learning regularities from data using spiking functions. Throughout this paper, the key to our theory is comparing the spiking behavior of the function on data samples and random samples. We say that a function $f$ discovers non-randomness from the data probability distribution, if the spiking frequency of $f$ on data samples differs significantly from that of $f$ on random samples. Then, taking the size of function $f$ into consideration, we claim that $f$ learns regularities from the data distribution if $f$ discovers non-randomness using a small size (or equivalently, a concise format). Finally, by referring to information theory, we propose that regularities can be regarded as a small amount of information encoding a large amount of information. Non-randomness is essentially valuable information in the data distribution.

After that, we apply multiple spiking functions to the same data distribution in order to learn the optimal regularities. We demonstrate that the optimal regularities shall capture the largest amount of information from the data distribution, and encode it into the smallest amount of information. We call the corresponding sequence of functions an optimal encoder to the data distribution. Numerical examples show that an explainable self-supervised learning system can be achieved by making the sequence of functions converge to an optimal encoder. That is, essentially, an explainable self-supervised learning system can be achieved by encoding the largest amount of information possible into the smallest amount of information possible.

In the future, realizing our theory by valid optimization algorithms and appropriate deep neural networks is the priority of our research.

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

## A  SUPPLEMENTAL DISCUSSIONS AND PROOF OF THEORIES

In this appendix, we will provide detailed definitions, discussions, hypotheses, and proofs related to Section 4. First, here is a more detailed introduction on metric, topology and Lebesgue measure regarding a vector space:

A **metric** $d$ on a vector space $\mathbf{X}$ is a function $d : \mathbf{X} \times \mathbf{X} \to \mathbb{R}$ that satisfies the following properties:

1. Non-negativity: $d(X, Y) \geq 0$ for any two vectors $X, Y \in \mathbf{X}$. Also, $d(X, Y) = 0$ if and only if $X = Y$.

2. Symmetry: $d(X, Y) = d(Y, X)$ for any $X, Y \in \mathbf{X}$.

3. Triangle Inequality: $d(X, Z) \leq d(X, Y) + d(Y, Z)$ for any $X, Y, Z \in \mathbf{X}$.

Then, the **Euclidean distance** on $\mathbf{X} = \mathbb{R}^m$ is defined to be $d(X, Y) = \sqrt{\sum_{j=1}^{m} (x_j - y_j)^2}$, where $X = (x_1, \cdots, x_m)$ and $Y = (y_1, \cdots, y_m)$ are two vectors in $\mathbb{R}^m$. The Euclidean distance on $\mathbf{X} = \mathbb{C}^m$ is defined to be $d(Z, W) = \sqrt{\sum_{j=1}^{m} |z_j - w_j|^2}$ for two vectors $Z = (z_1, \cdots, z_m)$ and $W = (w_1, \cdots, w_m)$ in $\mathbb{C}^m$. Here, $|z_j - w_j|$ is the modulus (or absolute value) (Stein & Shakarchi,

2010) of the complex number $z_j - w_j$. It is easy to prove that the Euclidean distance is a metric on $\mathbf{X} = \mathbb{R}^m$ and $\mathbf{X} = \mathbb{C}^m$.

With the defined metric $d$ on the vector space, we can define an open ball $B(X, r)$ around the vector $X \in \mathbf{X}$ as $B(X, r) = \{Y \in \mathbf{X} \mid d(X, Y) < r\}$. Then, a subset $\mathbf{E} \subset \mathbf{X}$ is said to be **open**, if for any $X \in \mathbf{E}$, there exists an $r > 0$ such that $B(X, r) \subset \mathbf{E}$. We can also construct the corresponding **topology** $\mathcal{T}$ on $\mathbf{S}$ by collecting all open sets in $\mathbf{X}$ (Sutherland, 2009). Then, given the domain $\mathbf{S} \subset \mathbf{X}$, a function $f : \mathbf{S} \to \mathbb{R}$ is said to be **continuous** around the vector $X \in \mathbf{S}$, if for any $\epsilon > 0$, there exists a $\delta > 0$ such that $d(X, Y) < \delta$ implies $|f(X) - f(Y)| < \epsilon$. Finally, $f$ is said to be continuous on $\mathbf{S}$ if it is continuous around every vector in $\mathbf{S}$.

Suppose $\mathbf{X} = \mathbb{R}^m$. Then, we define the *rectangular cuboid* $C$ on $\mathbb{R}^m$ to be a product $C = I_1 \times \cdots \times I_m$ of open intervals, with each open interval $I_j = (a_j, b_j)$ for $j = 1, \cdots, m$. Let $vol(C) = \prod_{j=1}^{m} |b_j - a_j|$ be the volume of $C$. Then, the Lebesgue outer measure $\lambda^*(\mathbf{E})$ for any subset $\mathbf{E} \subset \mathbb{R}^m$ is (where RC is the simplification of rectangular cuboids)

$$\lambda^*(\mathbf{E}) = \inf \left\{ \sum_{k=1}^{\infty} vol(C_k) : (C_k)_{k \in \mathbb{N}} \text{ is a countable sequence of RC with } \mathbf{E} \subset \bigcup_{k=1}^{\infty} C_k \right\}$$

where $\inf$ is the infimum (max lower bound) of the set of values.

Then, $\mathbf{E}$ is said to be **Lebesgue-measurable** (or simply measurable), if for any subset $\mathbf{A} \subset \mathbb{R}^m$, we have $\lambda^*(\mathbf{A}) = \lambda^*(\mathbf{A} \cap \mathbf{E}) + \lambda^*(\mathbf{A} \cap \mathbf{E}^c)$, where $\mathbf{E}^c = \mathbb{R}^m \backslash \mathbf{E}$ is the complement of $\mathbf{E}$ in $\mathbb{R}^m$. For any measurable subset $\mathbf{E} \subset \mathbb{R}^m$, its Lebesgue measure, denoted as $\lambda(\mathbf{E})$ or $|\mathbf{E}|$ in this paper, is defined to be its Lebesgue outer measure $\lambda^*(\mathbf{E})$. Finally, when $\mathbf{X} = \mathbb{C}^m$, the Lebesgue measure on $\mathbf{X}$ is defined with respect to the real space $\mathbb{R}^{2m}$ due to the isomorphism between $\mathbb{C}^m$ and $\mathbb{R}^{2m}$ (Downarowicz & Glasner, 2016).

Then, here is a more detailed discussion on non-measurable sets:

Non-measurable sets indeed exist within both real and complex vector spaces when considering the Lebesgue measure. A famous example is the Vitali set (Kharazishvili, 2011) defined on the closed interval $[0, 1]$: Given two real numbers $x, y \in [0, 1]$, we say that $x$ is *equivalent* to $y$ (denoted as $x \sim y$) if $x - y$ is a rational number. Then, for any $x \in [0, 1]$, its *equivalence class* is defined as $C_x = \{y \in [0, 1] | x \sim y\}$. In this way, the closed interval $[0, 1]$ can be partitioned into disjoint equivalence classes. Finally, from each equivalence class, we choose exactly one representative. This collection of representatives forms a Vitali set $V$ on $[0, 1]$. It can be proved that a Vitali set is not measurable under Lebesgue measure (Halmos, 2013). Intuitively, this means that there is no way to evaluate the volume of a Vitali set.

However, imagine that we have a function $f : [0, 1] \to \mathbb{R}$ that has its spiking region $\mathbf{S}_f$ to be a Vitali set. Intuitively, this will be extremely difficult: For any real number $x \in [0, 1]$, there is exactly one number $v$ in a Vitali set $V$ such that $x - v$ is rational. By definition, $v$ is the representative of $x$ in the equivalence class. This means that we can select at will uncountable many irrational numbers in $[0, 1]$ and obtain their corresponding representatives in $V$, which implies the extreme 'chaotic' of $V$. One can imagine how difficult it is for a function $f$ to map every number in a Vitali set to a positive value, while mapping all the other real numbers in $[0, 1]$ to negative values. In fact, we believe that it is impossible for $f$ to achieve this with a finite size $|f|$. Taking one step further, we believe that the same discussion can be applied to any non-measurable set on a finite-dimensional vector space, which leads to Hypothesis 1 (presented both here and in Section 3.2):

**Hypothesis 1.** *Suppose $\mathbf{X}$ is a finite-dimensional real or complex vector space, and suppose $\mathbf{S} \subset \mathbf{X}$ is a bounded sub-region in $\mathbf{X}$. Suppose $f : \mathbf{S} \to \mathbb{R}$ is a function defined on $\mathbf{S}$ with a finite size (i.e., there are finite adjustable parameters in $f$). Then, the spiking region $\mathbf{S}_f = \{X \in \mathbf{S} | f(X) > 0\}$ is always Lebesgue-measurable.*

After that, we provide the detailed proofs for the Lemmas and Theorems in the main paper.

**Lemma 1.** *Suppose $\mathbf{X}$ is a finite-dimensional real or complex vector space, and suppose $\mathbf{S} \subset \mathbf{X}$ is a bounded sub-region in $\mathbf{X}$. Suppose $f : \mathbf{S} \to \mathbb{R}$ is a continuous function defined on $\mathbf{S}$. Then, the spiking region $\mathbf{S}_f = \{X \in \mathbf{S} | f(X) > 0\}$ is always Lebesgue-measurable.*

*Proof.* Suppose $f(X) \leq 0$ for any $X \in \mathbf{S}$. Then, $\mathbf{S}_f = \emptyset$ (i.e., the empty set), which is measurable. Otherwise, suppose $\mathbf{S}_f \neq \emptyset$. For any $X \in \mathbf{S}_f$, we have $f(X) = r_X > 0$. Since $f$ is continuous on $\mathbf{S}$, there exists a $\delta > 0$ such that $d(X, Y) < \delta$ implies $|f(X) - f(Y)| < r_X/2$. This also means that $|f(Y)| > r_X/2 > 0$ when $d(X, Y) < \delta$. Hence, the open ball $B(X, \delta) = \{Y \in \mathbf{X} \mid d(X, Y) < \delta\}$ is entirely contained within $\mathbf{S}_f$. This means that $\mathbf{S}_f$ is open, which is Lebesgue-measurable (Nelson, 2015). $\square$

**Theorem 1.** *Suppose $\mathbf{X}$ is a finite-dimensional real or complex vector space, and suppose $\mathbf{S} \subset \mathbf{X}$ is a bounded sub-region in $\mathbf{X}$. Suppose we have the data probability distribution $\mathbf{P}$ defined on $\mathbf{S}$, with the probability density function to be $g(X)$. Furthermore, suppose there exists an upper bound $\Omega < \infty$, such that $g(X) \leq \Omega$ for any $X \in \mathbf{S}$. Finally, suppose we have the random distribution $\mathbf{P}'$ to be the uniform distribution defined on $\mathbf{S}$.*

*Then, for any function $f : \mathbf{S} \to \mathbb{R}$ with a finite size $|f|$, its theoretical spiking efficiency $SE_f$ obtained with respect to $\mathbf{P}$ and $\mathbf{P}'$ is bounded by $0 \leq SE_f \leq \Omega \cdot |\mathbf{S}| \cdot \log(\Omega \cdot |\mathbf{S}|)$, where $|\mathbf{S}|$ is the Lebesgue measure of data space $\mathbf{S}$.*

*Proof.* Suppose $\mathbf{S}_f$ is the spiking region of $f$. Since $|f|$ is finite, we know that $\mathbf{S}_f$ is measurable according to our hypothesis. And certainly, we require the data space $\mathbf{S}$ to be measurable in $\mathbf{X}$ with $|\mathbf{S}| > 0$. As we mentioned, $\mathbf{S}$ is bounded, indicating that $|\mathbf{S}| < \infty$. Then, suppose $f$ spikes on $M$ out of $N$ data samples $\{X_n\}_{n=1}^{N}$ generated by $\mathbf{P}$, as well as $M'$ out of $N$ random samples $\{X'_n\}_{n=1}^{N}$ generated by $\mathbf{P}'$. Also, suppose the probability density function of $\mathbf{P}'$ is $g'(X)$. By definition, we have $\int_{\mathbf{S}} g(X)\, dX = \int_{\mathbf{S}} g'(X)\, dX = 1$.

Since $\mathbf{P}'$ is the uniform distribution on $\mathbf{S}$, we have that $g'(X) = \frac{1}{|\mathbf{S}|}$ for any $X \in \mathbf{S}$. It is easy to see that $\Omega \geq \frac{1}{|\mathbf{S}|}$, otherwise we will get $\int_{\mathbf{S}} g(X)\, dX < 1$. This implies that $\psi = (\int_{\widehat{\mathbf{S}}} \Omega\, dX) / (\int_{\widehat{\mathbf{S}}} g'(X)\, dX) = (\int_{\widehat{\mathbf{S}}} \Omega\, dX) / (\int_{\widehat{\mathbf{S}}} \frac{1}{|\mathbf{S}|}\, dX) \geq 1$ for any region $\widehat{\mathbf{S}} \subset \mathbf{S}$ with $|\widehat{\mathbf{S}}| > 0$, which also means that $\log(\psi) \geq 0$.

Now, suppose $0 < |\mathbf{S}_f| < |\mathbf{S}|$, where $|\mathbf{S}_f|$ is the Lebesgue measure of the spiking region $\mathbf{S}_f$. This means that $0 < |\mathbf{S}_f^c| < |\mathbf{S}|$ as well, where $\mathbf{S}_f^c = \mathbf{S} \backslash \mathbf{S}_f$ is the complement of $\mathbf{S}_f$ in $\mathbf{S}$. Then, we have that:

$$SE_f = \lim_{N \to \infty} \left( \frac{M}{N} \log(\frac{M}{M'}) + \frac{N-M}{N} \log(\frac{N-M}{N-M'}) \right) \tag{16}$$

$$= \lim_{N \to \infty} \left( \frac{M}{N} \log(\frac{M/N}{M'/N}) + \frac{N-M}{N} \log(\frac{(N-M)/N}{(N-M')/N}) \right)$$

$$= \left( \int_{\mathbf{S}_f} g(X)\, dX \right) \cdot \log \left( \frac{\int_{\mathbf{S}_f} g(X)\, dX}{\int_{\mathbf{S}_f} g'(X)\, dX} \right) + \left( \int_{\mathbf{S}_f^c} g(X)\, dX \right) \cdot \log \left( \frac{\int_{\mathbf{S}_f^c} g(X)\, dX}{\int_{\mathbf{S}_f^c} g'(X)\, dX} \right) \tag{17}$$

$$\leq \left( \int_{\mathbf{S}_f} g(X)\, dX \right) \cdot \log \left( \frac{\int_{\mathbf{S}_f} \Omega\, dX}{\int_{\mathbf{S}_f} g'(X)\, dX} \right) + \left( \int_{\mathbf{S}_f^c} g(X)\, dX \right) \cdot \log \left( \frac{\int_{\mathbf{S}_f^c} \Omega\, dX}{\int_{\mathbf{S}_f^c} g'(X)\, dX} \right)$$

$$\leq \left( \int_{\mathbf{S}_f} \Omega\, dX \right) \cdot \log \left( \frac{\int_{\mathbf{S}_f} \Omega\, dX}{\int_{\mathbf{S}_f} g'(X)\, dX} \right) + \left( \int_{\mathbf{S}_f^c} \Omega\, dX \right) \cdot \log \left( \frac{\int_{\mathbf{S}_f^c} \Omega\, dX}{\int_{\mathbf{S}_f^c} g'(X)\, dX} \right)$$

$$= \Omega \cdot |\mathbf{S}_f| \cdot \log(\Omega \cdot |\mathbf{S}|) + \Omega \cdot (|\mathbf{S}| - |\mathbf{S}_f|) \cdot \log(\Omega \cdot |\mathbf{S}|)$$

$$= \Omega \cdot |\mathbf{S}| \cdot \log(\Omega \cdot |\mathbf{S}|) \tag{18}$$

Suppose $|\mathbf{S}_f| = 0$. Then, for each vector $X \in \mathbf{S}_f$, we construct an open ball $B(X, d_0) = \{Y \in \mathbf{S} \mid d(X, Y) < d_0\}$. Combining all the open balls for each $X \in \mathbf{S}_f$, we can get the open set $\mathbf{S}_1 = \bigcup_{X \in \mathbf{S}_f} B(X, d_0)$. Accordingly, if we reduce the radius from $d_0$ to $d_0/2$, we can get $\mathbf{S}_2 = \bigcup_{X \in \mathbf{S}_f} B(X, d_0/2)$. In general, we can get $\mathbf{S}_i = \bigcup_{X \in \mathbf{S}_f} B(X, d_0/2^{i-1})$ for $i \in \mathbb{N}$, with each $\mathbf{S}_i$ to be an open set (and hence measurable).

It is easy to see that $\mathbf{S}_{i+1} \subset \mathbf{S}_i$, $\mathbf{S}_f = \bigcap_{i=1}^{\infty} \mathbf{S}_i$, and $\mathbf{S}_f^c = \mathbf{S}\backslash(\bigcap_{i=1}^{\infty} \mathbf{S}_i) = \bigcup_{i=1}^{\infty}(\mathbf{S}\backslash\mathbf{S}_i) = \bigcup_{i=1}^{\infty} \mathbf{S}_i^c$. Also, by a routine derivation, we can get that there exists an $i_0 \in \mathbb{N}$, such that $0 < |\mathbf{S}_i| < |\mathbf{S}|$ and $0 < |\mathbf{S}_i^c| < |\mathbf{S}|$ will be true when $i \geq i_0$. Then, since integral over a zero-measure region is not directly defined, the definition of $SE_f$ when $|\mathbf{S}_f| = 0$ should be the limit of the integral over $\{\mathbf{S}_i\}_{i\in\mathbb{N}}$ as $i$ approaches $\infty$. That is,

$$SE_f = \lim_{i\to\infty} \left[ \left( \int_{\mathbf{S}_i} g(X)\,dX \right) \cdot \log\left( \frac{\int_{\mathbf{S}_i} g(X)\,dX}{\int_{\mathbf{S}_i} g'(X)\,dX} \right) + \left( \int_{\mathbf{S}_i^c} g(X)\,dX \right) \cdot \log\left( \frac{\int_{\mathbf{S}_i^c} g(X)\,dX}{\int_{\mathbf{S}_i^c} g'(X)\,dX} \right) \right] \quad (19)$$

$$\leq \lim_{i\to\infty} [\Omega \cdot |\mathbf{S}| \cdot \log(\Omega \cdot |\mathbf{S}|)] = \Omega \cdot |\mathbf{S}| \cdot \log(\Omega \cdot |\mathbf{S}|)$$

In fact, since the probability density function $g(X) \leq \Omega$ for any $X \in \mathbf{S}$, we can also prove by a routine derivation based on formula 19 that $SE_f = 0$ when $|\mathbf{S}_f| = 0$. Anyway, $SE_f \leq \Omega \cdot |\mathbf{S}| \cdot \log(\Omega \cdot |\mathbf{S}|)$ when $|\mathbf{S}_f| = 0$.

Suppose $|\mathbf{S}_f| = |\mathbf{S}|$ (in other words, we have $|\mathbf{S}_f^c| = 0$). Then, similarly, we can construct another sequence of open sets $\{\mathbf{S}_i\}_{i\in\mathbb{N}}$ such that $\mathbf{S}_{i+1} \subset \mathbf{S}_i$ and $\mathbf{S}_f^c = \bigcap_{i=1}^{\infty} \mathbf{S}_i$. Following exactly the same proving process, we can have that $SE_f \leq \Omega \cdot |\mathbf{S}| \cdot \log(\Omega \cdot |\mathbf{S}|)$ in this case.

Finally, as we mentioned in Section 3.1, $D_{KL}(\widehat{P}||\widehat{P}') \geq 0$ for any data samples $\{X_n\}_{n=1}^{N}$ and random samples $\{X_n'\}_{n=1}^{N}$. Then, we always have $SE_f = \lim_{N\to\infty} D_{KL}(\widehat{P}||\widehat{P}') \geq 0$. Hence, the proof is completed.

But before concluding this proof, we want to further discuss when $\mathbf{P}$ is not a 'regular' probability distribution. That is, there does not exist an upper bound $\Omega$ for the probability density function $g(X)$ of $\mathbf{P}$. The most simple case is that, there exists one singularity $X_{\mathbf{P}} \in \mathbf{S}$, such that $g(X_{\mathbf{P}}) = \infty$. For any sub-region $\widehat{\mathbf{S}} \subset \mathbf{S}$, we have $\int_{\widehat{\mathbf{S}}} g(X)\,dX = 1$ if $X_{\mathbf{P}} \in \widehat{\mathbf{S}}$. Otherwise $\int_{\widehat{\mathbf{S}}} g(X)\,dX = 0$. There can certainly be multiple singularities associated with $\mathbf{P}$. However, without loss of generality, we assume a unique singularity in $\mathbf{P}$.

Suppose $0 < |\mathbf{S}_f| < |\mathbf{S}|$ given a function $f : \mathbf{S} \to \mathbb{R}$, which also implies that $0 < |\mathbf{S}_f^c| < |\mathbf{S}|$. We can see that $X_{\mathbf{P}}$ must belong to either $\mathbf{S}_f$ or $\mathbf{S}_f^c$. We assume $X_{\mathbf{P}} \in \mathbf{S}_f$. Then, following the same derivation involved with formulas 17 and 18, we can have that $SE_f = \log(\frac{|\mathbf{S}|}{|\mathbf{S}_f|})$. Accordingly, by constructing a sequence of open sets $\{\mathbf{S}_i\}_{i\in\mathbb{N}}$ converging to $\mathbf{S}_f$, we can prove that when $|\mathbf{S}_f|$ converges to zero (i.e., $\mathbf{S}_f$ converges to $X_{\mathbf{P}}$), $SE_f = \log(\frac{|\mathbf{S}|}{|\mathbf{S}_f|})$ will converge to $\infty$. By assuming $X_{\mathbf{P}} \in \mathbf{S}_f^c$, we will have $SE_f = \log(\frac{|\mathbf{S}|}{|\mathbf{S}_f^c|}) = \log(\frac{|\mathbf{S}|}{|\mathbf{S}| - |\mathbf{S}_f|})$, and hence we can get the same result when $|\mathbf{S}_f^c|$ converges to zero (i.e., $\mathbf{S}_f^c$ converges to $X_{\mathbf{P}}$).

Intuitively, this means that if a zero-measure spiking region $\mathbf{S}_f$ contains the singularity of a data distribution $\mathbf{P}$, the corresponding function $f$ will then capture infinite amount of information from $\mathbf{P}$ by comparing $\mathbf{P}$ with a regular uniform distribution $\mathbf{P}'$.

At last, it seems that requiring the size $|f|$ of $f$ to be finite is redundant in our proof, which is in fact not true: Without $|f| < \infty$, we cannot guarantee that the spiking region $\mathbf{S}_f$ is Lebesgue measurable, which makes the derivation involving formulas 17 and 18 invalid. Hence, $|f| < \infty$ is necessary to our theorem. $\qquad \square$

After that, we provide the two hypotheses as mentioned in Section 3.3:

**Hypothesis 3.** *Suppose $\mathbf{X}$ is a finite-dimensional real or complex vector space, and suppose $\mathbf{S} \subset \mathbf{X}$ is a bounded sub-region in $\mathbf{X}$. Suppose there are the data probability distribution $\mathbf{P}$ and the uniform distribution $\mathbf{P}'$ defined on $\mathbf{S}$. Also, suppose the probability density function $g(X)$ of $\mathbf{P}$ is bounded by a finite number $\Omega$ (i.e., $\mathbf{P}$ is regular).*

*Then, for any sequence of finite-sized functions $\mathbf{f} = (f_1, \cdots, f_K)$, its spiking equivalence class $\mathcal{E}_{\mathbf{f}}$ contains infinite elements. That is, there are infinite sequences of finite-sized functions possessing the same theoretical spiking efficiency as $\mathbf{f}$.*

Providing a strict proof on this hypothesis is beyond the scope of this paper. But intuitively, suppose the spiking region of $\mathbf{f} = (f_1, \cdots, f_K)$ is $\mathbf{S}_{\mathbf{f}}$, and suppose the spiking region of $f_1$ (which is also

the independent spiking region) is $\mathbf{S}_{f_1}$. Suppose the dimension of the vector space $\mathbf{X}$ is $m$. Then, we find an $(m-1)$-dimensional hyperplane to divide $\mathbf{S}_{f_1} \subset \mathbf{X}$ into two pieces $\mathbf{S}_{f_1}^1$ and $\mathbf{S}_{f_1}^2$. Since $f_1$ has a finite size, we should be able to find two finite-sized functions $f_1^1$ and $f_1^2$, such that the spiking region of $f_1^1$ is $\mathbf{S}_{f_1}^1$ and that of $f_1^2$ is $\mathbf{S}_{f_1}^2$. Then, we can obtain a new sequence of functions $\widetilde{\mathbf{f}} = (f_1^1, f_1^2, f_2, \cdots, f_K)$ by replacing $f_1$ with $f_1^1$ and $f_1^2$. We can see that the spiking regions $\mathbf{S}_{\mathbf{f}}$ and $\mathbf{S}_{\widetilde{\mathbf{f}}}$ are coincide with each other, and hence the theoretical spiking efficiencies $SE_{\mathbf{f}} = SE_{\widetilde{\mathbf{f}}}$. This indicates that $\widetilde{\mathbf{f}} \in \mathcal{E}_{\mathbf{f}}$.

Since there are infinite ways to divide $\mathbf{S}_{f_1} \subset \mathbf{X}$ into $\mathbf{S}_{f_1}^1$ and $\mathbf{S}_{f_1}^2$, there are infinite new sequences of functions $\widetilde{\mathbf{f}}$ we can obtain. Hence, there are infinite elements in $\mathcal{E}_{\mathbf{f}}$. But again, this is not a strict proof.

**Hypothesis 4.** *Suppose $\mathbf{X}$ is a finite-dimensional real or complex vector space, and suppose $\mathbf{S} \subset \mathbf{X}$ is a bounded sub-region in $\mathbf{X}$. Suppose there are the data probability distribution $\mathbf{P}$ and the uniform distribution $\mathbf{P}'$ defined on $\mathbf{S}$. Also, suppose the probability density function $g(X)$ of $\mathbf{P}$ is bounded by a finite number $\Omega$ (i.e., $\mathbf{P}$ is regular).*

*With respect to $\mathbf{P}$ and $\mathbf{P}'$, suppose the most efficient class $\mathcal{E}_{\mathbf{P},\mathbf{P}'}^*$ is not empty. Then, every most efficient encoder $\mathbf{f}_{\mathbf{P},\mathbf{P}'}^* \in \mathcal{E}_{\mathbf{P},\mathbf{P}'}^*$ has exactly the same spiking region on $\mathbf{S}$.*

By looking into the example data distributions in Figure 2, we can intuitively agree to this hypothesis: Easy to understand that every most efficient encoder should have its spiking region cover exactly the data distributed region in each graph (namely, the two circles in the left graph, the overlapped diamonds in the middle graph, and the 15 squares in the right graph). But again, we do not aim at proving this hypothesis. Moreover, we hold the highest uncertainty to this hypothesis among all our proposed hypotheses in this paper.

Before ending this appendix section, we want to discuss again the reason for us to use uniform distribution $\mathbf{P}'$ as the random distribution throughout Section 3:

The optimal encoder $\mathbf{f}_{\mathbf{P},\mathbf{P}'}^\dagger$, the most efficient encoder $\mathbf{f}_{\mathbf{P},\mathbf{P}'}^*$, the most efficient class $\mathcal{E}_{\mathbf{P},\mathbf{P}'}^*$, and other concepts purposed by us share a common basis: The random distribution $\mathbf{P}'$ has to be the uniform distribution on the data space $\mathbf{S}$. Otherwise our theory will have to be adjusted into more complicated formats, since we need to consider the variance in the probability density function of $\mathbf{P}'$. However, we claim that even in that case, the essential definitions and hypotheses of our theory (regarding spiking equivalence, the most efficient encoder, and optimal encoders of different levels) will have similar formats. As a result, without loss of generality, we always assume $\mathbf{P}'$ to be the uniform distribution on the data space $\mathbf{S}$ in Section 3.

## B ENCODING MULTIPLE FUNCTIONS BY A MULTI-OUTPUT FUNCTION

This appendix introduces basic mathematical definitions and descriptions, which serves as a basis for our next appendix.

Suppose $\mathbf{X}$ is a finite-dimensional real or complex vector space, and suppose our data space $\mathbf{S}$ is a bounded sub-region in $\mathbf{X}$. Then, we define a multi-output function: Suppose $F : \mathbf{S} \to \mathbb{R}^H$ is a multi-output function mapping each vector $X \in \mathbf{S}$ into a real vector $(y_1, \cdots, y_H)$. Intuitively, we use each output head $y_h$ to 'mimic' a single-output function $y_h = f_h(X)$. We use $F|_h$ to denote each mimicked function $f_h : \mathbf{S} \to \mathbb{R}$ obtained in this way. That is, $(F|_1(X), \cdots, F|_H(X)) = (y_1, \cdots, y_H) = F(X)$ for any $X \in \mathbf{S}$.

We use $\mathbf{f}_F = (F|_1, \cdots, F|_H)$ to denote the sequence of single-output functions that is mimicked by each output head of $F$. Accordingly, we define the **independent spiking region** of each mimicked function $F|_h$ to be $\mathbf{S}_{F|_h}^{ind} = \{X \in \mathbf{S} \big| F|_h(X) > 0\}$. There is no necessary to consider the spiking region overlapping for each head $F|_h$. The independent spiking region of each head in a multi-output function will be enough for our discussion in this appendix section.

Following Section 3.1, we define the **size** of the multi-output function $F : \mathbf{S} \to \mathbb{R}^H$, denoted as $|F|$, to be the number of adjustable parameters in $F$, where a scalar parameter in $F$ is adjustable if its

value can be adjusted without changing the computational complexity of $F$. Then, we purpose the following hypothesis:

**Hypothesis 5.** *Suppose $\mathbf{X}$ is a finite-dimensional real or complex vector space, and suppose $\mathbf{S} \subset \mathbf{X}$ is a bounded sub-region in $\mathbf{X}$. Suppose $F : \mathbf{S} \to \mathbb{R}^H$ is a multi-output function with a finite size $|F|$, and suppose $\mathbf{f}_F = (F|_1, \cdots, F|_H)$ is the sequence of functions mimicked by the output heads in $F$.*

*Then, for each mimicked function $F|_h$, there always exists a real single-output function $f_h : \mathbf{S} \to \mathbb{R}$ with a finite size $|f_h|$, such that the (independent) spiking region $\mathbf{S}_{f_h}$ of $f_h$ exactly coincides with the independent spiking region $\mathbf{S}_{F|_h}^{ind}$ of $F|_h$. We call such a function $f_h : \mathbf{S} \to \mathbb{R}$ a **projection** of $F|_h$.*

We provide the following lemma that is necessary for further discussion:

**Lemma 2.** *Suppose $\mathbf{X}$ is a finite-dimensional real or complex vector space, and suppose $\mathbf{S} \subset \mathbf{X}$ is a bounded sub-region in $\mathbf{X}$. Suppose $f : \mathbf{S} \to \mathbb{R}$ is a function with a finite size, and suppose $\mathbf{S}_f$ is the spiking region of $f$.*

*Then, there exists a lower bound $\mathcal{L}_{\mathbf{S}_f}$ depending on $\mathbf{S}_f$, such that for any function $\widetilde{f} : \mathbf{S} \to \mathbb{R}$ with the same (coincided) spiking region as $\mathbf{S}_f$, the inequality $|\widetilde{f}| \geq \mathcal{L}_{\mathbf{S}_f}$ always holds true. Here, $|\widetilde{f}|$ is the size of $\widetilde{f}$.*

*Finally, there always exists a function $f^\dagger : \mathbf{S} \to \mathbb{R}$, such that the spiking region of $f^\dagger$ coincides with $\mathbf{S}_f$, and $|f^\dagger| = \mathcal{L}_{\mathbf{S}_f}$. We call $f^\dagger$ an **optimal function** to $\mathbf{S}_f$.*

*Proof.* The size $|f|$ of function $f$ is defined to be the number of adjustable parameters in $f$, which is a finite positive integer as supposed in this lemma. If a function $\widetilde{f} : \mathbf{S} \to \mathbb{R}$ has the spiking region coincided to $\mathbf{S}_f$ and is of a smaller size, then the possible values of $|\widetilde{f}|$ are constrained to the finite set $\{1, 2, \cdots, |f|\}$. As a result, both the lower bound $\mathcal{L}_{\mathbf{S}_f}$ and the size $|f^\dagger|$ of the optimal function $f^\dagger$ will be the minimum value achievable in $\{1, 2, \cdots, |f|\}$. $\square$

Note that there can exist multiple optimal functions to the same spiking region. Intuitively, an optimal function $f^\dagger$ to the spiking region $\mathbf{S}_f$ encodes the minimum amount of information required to describe $\mathbf{S}_f$ unambiguously. Then, the following lemma shows that the definitions on optimal encoders and optimal functions are self-consistent:

**Lemma 3.** *Suppose $\mathbf{X}$ is a finite-dimensional real or complex vector space, and suppose $\mathbf{S} \subset \mathbf{X}$ is a bounded sub-region in $\mathbf{X}$. Suppose there are the data probability distribution $\mathbf{P}$ and the uniform distribution $\mathbf{P}'$ defined on $\mathbf{S}$. Also, suppose the probability density function $g(X)$ of $\mathbf{P}$ is bounded by a finite number $\Omega$ (i.e., $\mathbf{P}$ is regular). Finally, suppose an optimal encoder $\mathbf{f}_{\mathbf{P}, \mathbf{P}'}^\dagger = (f_1^\dagger, \cdots, f_{K\dagger}^\dagger)$ exists regarding $\mathbf{P}$ and $\mathbf{P}'$.*

*Then, each function $f_k^\dagger$ in the optimal encoder is also an optimal function to its own independent spiking region $\mathbf{S}_{f_k^\dagger}^{ind}$.*

*Proof.* Assume that the statement is false for one function $f_k^\dagger$ in $(f_1^\dagger, \cdots, f_{K\dagger}^\dagger)$. Then, there exists another function $\widetilde{f} : \mathbf{S} \to \mathbb{R}$ whose spiking region coincides with $\mathbf{S}_{f_k^\dagger}^{ind}$, and has its size $|\widetilde{f}| < |f_k^\dagger|$. Then, if we replace $f_k^\dagger$ by $\widetilde{f}$ in the sequence of functions $(f_1^\dagger, \cdots, f_{K\dagger}^\dagger)$, $\widetilde{f}$ will produce the same spiking region as $\mathbf{S}_{f_k^\dagger} = \mathbf{S}_{f_k^\dagger}^{ind} \backslash (\cup_{i=1}^{k-1} \mathbf{S}_{f_i^\dagger}^{ind})$, and also produce the same theoretical spiking efficiency as $SE_{f_k^\dagger}$. But then, the theoretical ability of $\widetilde{f}$ will be $A_{\widetilde{f}} = SE_{f_k^\dagger} \cdot |\widetilde{f}|^{-1} > A_{f_k^\dagger}$, while the theoretical abilities of other functions in the sequence are the same. Hence, replacing $f_k^\dagger$ by $\widetilde{f}$, we will obtain a new sequence of functions with a larger theoretical ability than $A_{\mathbf{f}_{\mathbf{P}, \mathbf{P}'}^\dagger}$, which contradicts our definition on an optimal encoder. Hence, the statement is true. $\square$

Now, we combine Lemma 2 and Hypothesis 5 in our discussion: Suppose $\mathbf{f}_F = (F|_1, \cdots, F|_H)$ is the sequence of functions mimicked by each head in the multi-output function $F : \mathbf{S} \to \mathbb{R}^H$, and suppose the independent spiking region of each mimicked function $F|_h$ is $\mathbf{S}_{F|_h}^{ind}$. Since there always

exists a projection (finite-sized single-output function) $f_h : \mathbf{S} \to \mathbb{R}$ whose spiking region coincides with $\mathbf{S}_{F|_h}^{ind}$, we can see that there always exists an optimal function $f_h^\dagger : \mathbf{S} \to \mathbb{R}$ with respect to $\mathbf{S}_{F|_h}^{ind}$. We call such an optimal function $f_h^\dagger$ an **optimal projection** of $F|_h$.

Then, what is the relationship between the size $|F|$ of the multi-output function $F$, and the size summation $\sum_{h=1}^{H} |f_h^\dagger|$ of optimal projections regarding all the output heads of $F$? In the specific case as shown in Figure 3, the summation $\sum_{h=1}^{H} |f_h^\dagger|$ can be much larger than $|F|$:

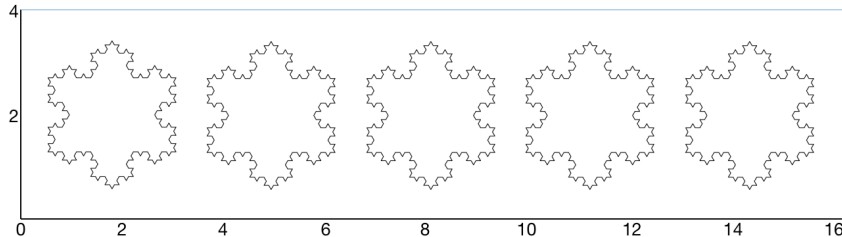

Figure 3: A specific case where the size summation $\sum_{h=1}^{H} |f_h^\dagger|$ of optimal projections is much larger than the size $|F|$.

In Figure 3, we have the data space $\mathbf{S} = \{x, y | 0 \le x \le 16, 0 \le y \le 4\}$ to be a sub-region in $\mathbb{R}^2$. Suppose the single-output function $f^\dagger : \mathbf{S} \to \mathbb{R}$ is an optimal function to a third-iteration Koch snowflake (Lapidus & Pang, 1995). Then, for $(x, y) \in \mathbf{S}$, we define the multi-output function $F : \mathbf{S} \to \mathbb{R}^5$ to be $F(x, y) = (f^\dagger(x, y), f^\dagger(x - \beta, y), f^\dagger(x - 2\beta, y), f^\dagger(x - 3\beta, y), f^\dagger(x - 4\beta, y))$. With $\beta \approx 3$, we can have the independent spiking regions of the mimicked functions in $\mathbf{f}_F = (F|_1, F|_2, F|_3, F|_4, F|_5)$ to be the five Koch snowflakes as shown in Figure 3. With $f_h^\dagger$ to be the optimal projection of each $F|_h$, we have that $\frac{\sum_{h=1}^{5} |f_h^\dagger|}{|F|} = \frac{5|f^\dagger|}{|f^\dagger| + 4}$. In fact, for a general head number $H$ in this example, we have that $\frac{\sum_{h=1}^{H} |f_h^\dagger|}{|F|} = \frac{H|f^\dagger|}{|f^\dagger| + H - 1}$, which converges to $|f^\dagger|$ when $H$ is large enough. This shows that the ratio $\frac{\sum_{h=1}^{H} |f_h^\dagger|}{|F|}$ is not bounded in this example.

However, intuitively, this kind of example is rare. We believe that in general, the independent spiking region of each mimicked function $F|_h$ in $\mathbf{f}_F = (F|_1, \cdots, F|_H)$ is likely to be 'irrelevant' to each other:

**Definition 5.** *Suppose $\mathbf{X}$ is a finite-dimensional real or complex vector space, and suppose $\mathbf{S} \subset \mathbf{X}$ is a bounded sub-region in $\mathbf{X}$. Suppose $\mathbf{S}_1 \subset \mathbf{S}$ and $\mathbf{S}_2 \subset \mathbf{S}$ are two sub-regions in $\mathbf{S}$. Suppose $f_1^\dagger, f_2^\dagger : \mathbf{S} \to \mathbb{R}$ are the optimal functions to $\mathbf{S}_1$ and $\mathbf{S}_2$, respectively.*

*We say that a mapping $\mathcal{M} : \mathbf{S} \to \mathbf{S}$ is a **bijection** from $\mathbf{S}_1$ to $\mathbf{S}_2$, if for any vector $Y \in \mathbf{S}_2$, there exists a unique vector $X \in \mathbf{S}_1$ such that $Y = \mathcal{M}(X)$. We define the **size** of $\mathcal{M}$, denoted as $|\mathcal{M}|$, to be the number of adjustable parameters in $\mathcal{M}$, where a scalar parameter in $\mathcal{M}$ is adjustable if its value can be adjusted without changing the computational complexity of $\mathcal{M}$.*

*Then, we say that $\mathbf{S}_2$ is **irrelevant** to $\mathbf{S}_1$, if for any bijection $\mathcal{M}$ from $\mathbf{S}_1$ to $\mathbf{S}_2$, we always have $|\mathcal{M}| \ge |f_2^\dagger|$.*

Again, the size of a mapping $\mathcal{M}$ is defined to be the number of adjustable parameters in $\mathcal{M}$, which is essentially equivalent to the minimum amount of information required to describe $\mathcal{M}$ unambiguously. And again, the size of an optimal encoder $f^\dagger$ represents the minimum amount of information required to describe its spiking region $\mathbf{S}_{f^\dagger}$ unambiguously. So, intuitively, a sub-region $\mathbf{S}_2 \subset \mathbf{S}$ is irrelevant to $\mathbf{S}_1 \subset \mathbf{S}$, if knowing the information required to describe $\mathbf{S}_1$ will not reduce the amount of information required to describe $\mathbf{S}_2$.

There is another interesting hypothesis: For two sub-regions $\mathbf{S}_1 \subset \mathbf{S}$ and $\mathbf{S}_2 \subset \mathbf{S}$, $\mathbf{S}_2$ is irrelevant to $\mathbf{S}_1$ if and only if $\mathbf{S}_1$ is irrelevant to $\mathbf{S}_2$. Once again, we do not dive deep into proving this intuitively correct hypothesis. We say that two sub-regions $\mathbf{S}_1$ and $\mathbf{S}_2$ are **mutually irrelevant**, if $\mathbf{S}_2$ is irrelevant to $\mathbf{S}_1$ and $\mathbf{S}_1$ is irrelevant to $\mathbf{S}_2$.

Applying this to our multi-output function approach, we hope to provide the following hypothesis:

**Hypothesis 6.** *Suppose $\mathbf{X}$ is a finite-dimensional real or complex vector space, and suppose $\mathbf{S} \subset \mathbf{X}$ is a bounded sub-region in $\mathbf{X}$. Suppose $F : \mathbf{S} \to \mathbb{R}^H$ is a multi-output function with a finite size $|F|$, and suppose $\mathbf{f}_F = (F|_1, \cdots, F|_H)$ is the sequence of functions mimicked by the output heads in $F$. Suppose $f_h^{\dagger} : \mathbf{S} \to \mathbb{R}$ is an optimal projection of each mimicked function $F|_h$ in $\mathbf{f}_F$, whose size is $|f_h^{\dagger}| < \infty$.*

*Furthermore, suppose the independent spiking regions $\mathbf{S}_{F|_h}^{ind}$ and $\mathbf{S}_{F|_j}^{ind}$ of $F|_h$ and $F|_j$ are mutually irrelevant, when $h \neq j$. Then, we have that $\sum_{h=1}^{H} |f_h^{\dagger}| \leq |F|$.*

Intuitively, this hypothesis implies that given a multi-output function $F$, when the independent spiking region regarding each of its output head is mutually irrelevant to each other, there is no way for $F$ to further compress the information required to specify these independent spiking regions.

This hypothesis does not contradict to our main theory: A function $f$ should encode a large amount of information from the data distribution into a small amount of information (i.e. the parameters in $f$), when $f$ discovers non-randomness from the data distribution. On contrast, mutually irrelevant spiking regions do not contain non-randomness. In other words, all the information required to describe mutually irrelevant spiking regions are totally random, which henceforth cannot be further encoded or compressed into a smaller amount of information.

After reading this appendix section, one may be able to better understand the next appendix section, in which we always assume two output heads in each multi-output function $F_1, \cdots, F_K$. In the next appendix section, we discusses our goal for the independent spiking region of the second head $(F_k|_2)$ in each bi-output function $F_k$ to converge to fixed random samples in the data space, while the independent spiking region of the first head $(F_k|_1)$ converges to optimized data regions.

If this goal is achieved, the independent spiking regions of $F_k|_1$ and $F_k|_2$ in each bi-output function $F_k$ should become irrelevant to each other. By a routine analysis, we can see that the size of the optimal projection regarding $F_k|_2$ is at least $m \cdot L_k'$, where $m$ is the dimension of the vector space, and $L_k'$ is the number of fixed random vectors making $F_k|_2$ spike (more details can be found in the next Appendix sub-section C.1). Therefore, the size of the optimal projection regarding $F_k|_1$ can be estimated by $|F_k| - mL_k'$ according to Hypothesis 6.

As mentioned at the end of Section 3.3, there can exist a regular probability distribution without a most efficient encoder. We build such a counter example at the end of this appendix. We refer to Figure 3 again: Imagining that we have a data distribution $\mathbf{P}$ which is uniformly distributed in a full Koch snowflake. Also, we assume that the data space $\mathbf{S}$ is large enough comparing to the Koch snowflake, and $\mathbf{P}'$ is the uniform distribution on $\mathbf{S}$. It is easy to see that $\mathbf{P}$ is regular (i.e., the probability density function of $\mathbf{P}$ is bounded). Suppose $f : \mathbf{S} \to \mathbb{R}$ is a finite-sized function whose spiking region coincides with the $L$-level iteration of the Koch snowflake in which $\mathbf{P}$ distributed.

Then, suppose $\widetilde{f} : \mathbf{S} \to \mathbb{R}$ is another finite-sized function whose spiking region coincides with the $(L+1)$-level iteration of such Koch snowflake. It is easy to see that the spiking region $\mathbf{S}_f \subset \mathbf{S}_{\widetilde{f}}$ (Lapidus & Pang, 1995). Since the data space $\mathbf{S}$ is large enough and $\mathbf{P}$ is uniformly distributed within the full Koch snowflake, by a routine analysis involving formulas 6 and 7, we can get the theoretical spiking efficiency $SE_f \leq SE_{\widetilde{f}}$ with respect to $\mathbf{P}$ and $\mathbf{P}'$. This implies that the higher iteration level the spiking region coincides with the data distribution Koch snowflake, the larger the theoretical spiking efficiency of the function will be. However, it is not possible for a function with a finite-size to possess a spiking region coinciding with the full Koch snowflake: It will need infinite amount of parameters to represent a full fractal (Mandelbrot, 1989). As a result, there is no most efficient encoder with respect to $\mathbf{P}$ and $\mathbf{P}'$ in this example, leading to $\mathcal{E}_{\mathbf{P},\mathbf{P}'}^* = \emptyset$.

In the next appendix, we will propose our designed machine learning pipeline, which aims at discovering optimal encoders for a given data probability distribution in practice.

# C    APPLYING MULTIPLE BI-OUTPUT FUNCTIONS TO APPROACH TO OPTIMAL ENCODERS IN PRACTICE

We introduce our designed machine learning approach in this section. This approach aims at discovering optimal encoders for a given data probability distribution in practice. The first part describes the algorithms and formulas related to our designed approach. The second part describes a pipeline to implement our designed approach in a layer-wise manner on a given image dataset. Again, we only provide our designed approach and pipeline here. There is no actual realization or experimental result involved.

## C.1    DESIGNED MACHINE LEARNING APPROACH

Once again, suppose $\mathbf{X}$ is a finite-dimensional real or complex vector space, and suppose our data space $\mathbf{S}$ is a bounded sub-region in $\mathbf{X}$. Suppose we have the dataset $\mathcal{D} = \{X_1, X_2, \ldots, X_{N_{\mathcal{D}}}\}$, with each sample vector $X_n \in \mathbf{S}$. We use $\mathbf{P}_{\mathcal{D}}$ to denote the data probability distribution generating $\mathcal{D}$. We define $\mathbf{P}'$ to be the uniform distribution on $\mathbf{S}$. In practice, we always assume $\mathbf{P}_{\mathcal{D}}$ to be regular (i.e., its probability density function is bounded).

In order to approach to an optimal encoder with respect to $\mathbf{P}_{\mathcal{D}}$ and $\mathbf{P}'$, we create a bi-output function $F : \mathbf{S} \to \mathbb{R}^2$ that maps a vector $X \in \mathbf{S}$ to a 2D real vector $Y = (y_1, y_2)$. In practice, $F$ should be a deep neural network with developed architectures, such as a multi-layer convolutional neural network (CNN) (Li et al., 2021) combined with non-linear activation functions (like a ReLU function)(Nair & Hinton, 2010; Agarap, 2018), layer normalization (Ba et al., 2016), and fully connected layers. The output of $F$ should consist of two scalars, each of which is produced by a hyperbolic tangent function ('tanh') (Lau & Lim, 2018) to restrict the scalar between -1 and 1.

We define the **size** of the bi-output function $F$, denoted as $|F|$, to be the number of adjustable parameters in $F$. Here, a scalar parameter in $F$ is adjustable if its value can be adjusted without changing the computational complexity of $F$.

Then, suppose we generate $L$ random samples, denoted as $\mathcal{D}'_{\text{fix}} = \{X'_1, \cdots, X'_L\}$, from the uniform distribution $\mathbf{P}'$, and then fix these samples. Given the bi-output function $F$, suppose the second head of $F$ spikes on $L'$ random samples in $\mathcal{D}'_{\text{fix}}$. That is, we have $\mathcal{D}'_{\text{fix},F} \subset \mathcal{D}'_{\text{fix}}$ containing $L'$ random samples, such that for each $X' \in \mathcal{D}'_{\text{fix},F}$, we have $y_2 > 0$ in $(y_1, y_2) = F(X')$. Ideally, we desire the second head of $F$ to only spike on these $L'$ fixed random samples in $\mathcal{D}'_{\text{fix},F}$, or spike on the random samples that are extremely close to each $X' \in \mathcal{D}'_{\text{fix},F}$. That is, we hope the spiking region regarding the second head of $F$ to consist of very tiny regions around each $X' \in \mathcal{D}'_{\text{fix},F}$, as shown in Figure 4.

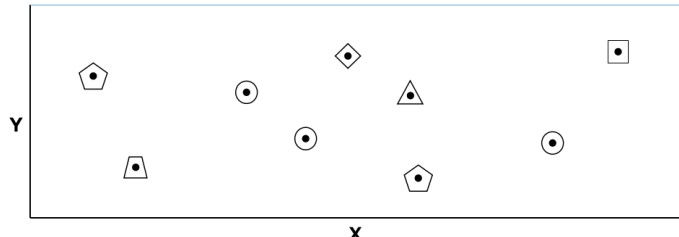

Figure 4: Suppose the data space $\mathbf{S} \subset \mathbb{R}^2$, and suppose the second head of $F$ spikes on fixed random samples (points in the figure) in $\mathcal{D}'_{\text{fix},F}$. Then, the desired spiking region regarding the second head of $F$ should be the union of the circles, squares and polygons in the figure, converging to each random sample in $\mathcal{D}'_{\text{fix},F}$.

Since the random samples in $\mathcal{D}'_{\text{fix}} = \{X'_1, \cdots, X'_L\}$ are independent and identically distributed (i.i.d), so are the random samples in $\mathcal{D}'_{\text{fix},F}$. Hence, there is no way to further encode or compress the information required to describe these 'randomly distributed random samples' in $\mathcal{D}'_{\text{fix},F}$. This means that the bi-output function $F$ has to record in its parameters the full amount of information describing $\mathcal{D}'_{\text{fix},F}$, in order to have the spiking region of its second head converging to each $X' \in \mathcal{D}'_{\text{fix},F}$. We provide a theoretical analysis regarding this issue in the previous appendix section.

Suppose the vector space is $\mathbf{X} = \mathbb{R}^m$. Then, it requires in total $m \cdot L'$ parameters to record the information describing all $L'$ random samples in $\mathcal{D}'_{\text{fix},F}$. When $\mathbf{X} = \mathbb{C}^m$, the amount of required parameters becomes $2m \cdot L'$. But without lose of generality, we assume $\mathbf{X} = \mathbb{R}^m$. The way for $F$ to record such information may be implicit due to the nature of deep neural networks (Markatopoulou et al., 2018). But in whatever way, $F$ has to consume equivalent to $mL'$ parameters to record such information. Then, if we desire the first head of the bi-output function $F$ to have a different spiking behavior, the available amount of parameters is at most $|F| - mL'$.

We use $F|_1$ and $F|_2$ to denote the single-output function 'mimicked' by the first and second head of $F$, respectively. That is, $(F|_1(X), F|_2(X)) = (y_1, y_2) = F(X)$ for any $X \in \mathbf{S}$. In this way, we can obtain two 'mimicked' functions $F|_1, F|_2 : \mathbf{S} \to \mathbb{R}$. A more detailed discussion is provided the previous appendix section. But intuitively, it is not difficult to understand that the size of $F|_1$ can be estimated by $|F| - mL'$, indicating implicitly the amount of adjustable parameters in $F$ that is available for the first head.

Suppose we select $N$ data samples $\{X_n\}_{n=1}^N$ from dataset $\mathcal{D}$, and generate $N$ random samples $\{X'_n\}_{n=1}^N$ by the uniform distribution $\mathbf{P}'$. Note that $\{X_n\}_{n=1}^N$ is independent with $\mathcal{D}'_{\text{fix}}$. According to the above discussion, $F|_2$ should hardly spike on any sample in $\{X_n\}_{n=1}^N$ or $\{X'_n\}_{n=1}^N$. Otherwise, the amount of information required to describe the spiking region of $F|_2$ may likely be less than $mL'$, which invalidates our design. So, we assume that $F|_2$ spikes on $M_2$ data samples in $\{X_n\}_{n=1}^N$ and $M'_2$ random samples in $\{X'_n\}_{n=1}^N$. We use $L' - \lambda(M_2 + M'_2)$ to measure the 'valid' spikings made by $F|_2$ on the fixed random samples in $\mathcal{D}'_{\text{fix}}$, with $\lambda$ to be far larger than one (say, $\lambda = 50$). During optimization in practice, $M_2 + M'_2$ should be reduced by high pressure from $\lambda$, which will then keep as many valid spikings as possible for $F|_2$. Accordingly, the size of $F|_1$ is estimated by $|F| - m(L' - \lambda(M_2 + M'_2))$.

Suppose we have $K$ bi-output functions $F_1, \cdots, F_K : \mathbf{S} \to \mathbb{R}^2$, which are arranged in the sequence $\mathbf{f} = (F_1, \cdots, F_K)$. Again, each $F_k$ is a deep neural network in practice. We use $F_k|_1$ and $F_k|_2$ to denote the single-output function mimicked by the first and second head of $F_k$, respectively. We perform each $F_k$ on the same data samples $\{X_n\}_{n=1}^N$ selected from $\mathcal{D}$, the same random samples $\{X'_n\}_{n=1}^N$ generated by $\mathbf{P}'$, and the $L$ fixed random samples in $\mathcal{D}'_{\text{fix}}$.

We use $\mathbf{f}_{\text{mimic}} = (F_1|_1, \cdots, F_K|_1)$ to denote the sequence of single-output functions mimicked by the first head of $F_1, \cdots, F_K$. Then, $\mathbf{f}_{\text{mimic}}$ will be the sequence of functions we work with. We now discuss how to obtain the observed spiking efficiency and observed ability of $\mathbf{f}_{\text{mimic}}$.

Given each $F_k|_1$ in $\mathbf{f}_{\text{mimic}}$, suppose there are $M_{k,1}$ data samples in $\{X_n\}_{n=1}^N$ and $M'_{k,1}$ random samples in $\{X'_n\}_{n=1}^N$ that make $F_k|_1$ spike, but do not make $F_1|_1, \cdots, F_{k-1}|_1$ spike. Accordingly, with respect to $\mathbf{P}_\mathcal{D}$ and $\mathbf{P}'$, the observed spiking efficiency of $F_k|_1$, denoted as $\widehat{SE}_{F_k|_1}$, can be calculated as:

$$\widehat{SE}_{F_k|_1} = \frac{M_{k,1}}{N}\log(\frac{M_{k,1}+\alpha}{M'_{k,1}+\alpha}) + \frac{N-M_{k,1}}{N}\log(\frac{N-M_{k,1}+\alpha}{N-M'_{k,1}+\alpha}). \tag{20}$$

Suppose $M_{\mathbf{f}_{\text{mimic}}} = \sum_{k=1}^K M_{k,1}$ and $M'_{\mathbf{f}_{\text{mimic}}} = \sum_{k=1}^K M'_{k,1}$. Similar to Section 3.3, we know that $M_{\mathbf{f}_{\text{mimic}}}$ is the total number of data samples in $\{X_n\}_{n=1}^N$ that make at least one mimicked function in $\mathbf{f}_{\text{mimic}} = (F_1|_1, \cdots, F_K|_1)$ to spike. Also, $M'_{\mathbf{f}_{\text{mimic}}}$ is the total number of random samples in $\{X'_n\}_{n=1}^N$ that make at least one mimicked function in $\mathbf{f}_{\text{mimic}}$ to spike. Then, we can get the observed spiking efficiency of $\mathbf{f}_{\text{mimic}}$, denoted as $\widehat{SE}_{\mathbf{f}_{\text{mimic}}}$, as:

$$\widehat{SE}_{\mathbf{f}_{\text{mimic}}} = \frac{M_{\mathbf{f}_{\text{mimic}}}}{N}\log(\frac{M_{\mathbf{f}_{\text{mimic}}}+\alpha}{M'_{\mathbf{f}_{\text{mimic}}}+\alpha}) + \frac{N-M_{\mathbf{f}_{\text{mimic}}}}{N}\log(\frac{N-M_{\mathbf{f}_{\text{mimic}}}+\alpha}{N-M'_{\mathbf{f}_{\text{mimic}}}+\alpha}). \tag{21}$$

Also, for each bi-output function $F_k$ in $\mathbf{f} = (F_1, \cdots, F_K)$, suppose there are $M_{k,2}$ data samples in $\{X_n\}_{n=1}^N$ and $M'_{k,2}$ random samples in $\{X'_n\}_{n=1}^N$ that make $F_k|_2$ spike. Note that there are no shared weights among different $F_k$. So, there is no further restriction from $F_1, \cdots, F_{k-1}$ upon $M_{k,2}$ and $M'_{k,2}$. Also, suppose there are $L'_k$ fixed random samples in $\mathcal{D}'_{\text{fix}}$ that make $F_k|_2$ spike. According to our previous discussion, the size of $F_k|_1$ can be estimated by $|F_k|_1| = |F_k| - m(L'_k - \lambda(M_{k,2} + M'_{k,2}))$. Then, the observed ability of each mimicked function $F_k|_1$ in $\mathbf{f}_{\text{mimic}} = (F_1|_1, \cdots, F_K|_1)$ can be estimated by $\widehat{A}_{F_k|_1} = \widehat{SE}_{F_k|_1} \cdot |F_k|_1|^{-1}$.

Finally, the observed ability of $\mathbf{f}_{\text{mimic}}$ can be estimated by $\widehat{A}_{\mathbf{f}_{\text{mimic}}} = \sum_{k=1}^{K} \widehat{A}_{F_k|_1} = \sum_{k=1}^{K} [\widehat{SE}_{F_k|_1} \cdot |F_k|_1|^{-1}]$. Based on our discussion at the end of Section 3.4, the $\mathcal{O}$bjective of learning is to maximize both $\widehat{SE}_{\mathbf{f}_{\text{mimic}}}$ and $\widehat{A}_{\mathbf{f}_{\text{mimic}}}$. That is, we want to maximize:

$$
\mathcal{O}_{\mathbf{f}_{\text{mimic}}} = \lambda_1 \cdot \left( \frac{M_{\mathbf{f}_{\text{mimic}}}}{N} \log\left(\frac{M_{\mathbf{f}_{\text{mimic}}} + \alpha}{M'_{\mathbf{f}_{\text{mimic}}} + \alpha}\right) + \frac{N - M_{\mathbf{f}_{\text{mimic}}}}{N} \log\left(\frac{N - M_{\mathbf{f}_{\text{mimic}}} + \alpha}{N - M'_{\mathbf{f}_{\text{mimic}}} + \alpha}\right) \right) +
$$

$$
\lambda_2 \cdot \sum_{k=1}^{K} \left[ \left( \frac{M_{k,1}}{N} \log\left(\frac{M_{k,1} + \alpha}{M'_{k,1} + \alpha}\right) + \frac{N - M_{k,1}}{N} \log\left(\frac{N - M_{k,1} + \alpha}{N - M'_{k,1} + \alpha}\right) \right) \cdot \frac{1}{|F_k| - m(L'_k - \lambda(M_{k,2} + M'_{k,2}))} \right],
$$

(22)

where $\lambda_1$, $\lambda_2$ and $\lambda$ are pre-defined hyper-parameters. In some cases, increasing $\widehat{SE}_{\mathbf{f}_{\text{mimic}}}$ by a small margin may require the mimicked functions in $\mathbf{f}_{\text{mimic}}$ to enlarge their sizes significantly, which reduces $\widehat{A}_{\mathbf{f}_{\text{mimic}}}$ and ultimately reduces $\mathcal{O}_{\mathbf{f}_{\text{mimic}}}$. Intuitively, the ratio $\lambda_1/\lambda_2$ can be viewed as the tolerance for achieving increased spiking efficiency through size expansion. So, appropriately choosing $\lambda_1$ and $\lambda_2$ becomes important in practice.

Here, we summarize our designed approach, which to some extent can be regarded as an unsupervised feature extraction method (Ghahramani, 2003):

Suppose $\mathbf{X} = \mathbb{R}^m$ is the vector space, and $\mathbf{S} \subset \mathbf{X}$ is our bounded data space. Suppose we have the dataset $\mathcal{D} = \{X_1, X_2, \ldots, X_{N_\mathcal{D}}\}$, with each vector $X_n \in \mathbf{S}$. We assume it is the data probability distribution $\mathbf{P}_\mathcal{D}$ that generates $\mathcal{D}$. Also, we define the uniform distribution $\mathbf{P}'$ on $\mathbf{S}$. Then, we use $\mathbf{P}'$ to generate and fix $L$ random samples in $\mathcal{D}'_{\text{fix}} = \{X'_1, \cdots, X'_L\}$, with $L$ to be large enough. Finally, we initialize $K$ deep neural networks $F_1, \cdots, F_K$. Each $F_k$ takes a vector $X \in \mathbf{S}$ as its input and provides a 2-dimensional output $(y_1, y_2)$. We put the neural networks in a sequence $\mathbf{f} = (F_1, \cdots, F_K)$, and obtain the sequence of their first head mimicked functions $\mathbf{f}_{\text{mimic}} = (F_1|_1, \cdots, F_K|_1)$.

In each training (learning) step, we randomly select $N$ data samples $\{X_n\}_{n=1}^{N}$ from $\mathcal{D}$ and generate $N$ random samples $\{X'_n\}_{n=1}^{N}$ by $\mathbf{P}'$, with $N$ to be large enough. For each $F_k$ in $(F_1, \cdots, F_K)$, we obtain its spiking scores $M_{k,1}$, $M'_{k,1}$, $M_{k,2}$, $M'_{k,2}$ and $L'_k$ according to the above discussion. Then, we shall use an optimization algorithm to maximize $\mathcal{O}_{\mathbf{f}_{\text{mimic}}}$ in formula 22 with respect to $\{M_{k,1}, M'_{k,1}, M_{k,2}, M'_{k,2}, L'_k\}_{k=1}^{K}$, $M_{\mathbf{f}_{\text{mimic}}} = \sum_{k=1}^{K} M_{k,1}$ and $M'_{\mathbf{f}_{\text{mimic}}} = \sum_{k=1}^{K} M'_{k,1}$.

We repeat this process with new $\{X_n\}_{n=1}^{N}$ and $\{X'_n\}_{n=1}^{N}$ in each training step until we are satisfied, which makes $\mathbf{f}_{\text{mimic}} = (F_1|_1, \cdots, F_K|_1)$ converge to a potential optimal encoder with respect to $\mathbf{P}_\mathcal{D}$ and $\mathbf{P}'$, denoted as $\mathbf{f}^\dagger_{\mathbf{P}_\mathcal{D}, \mathbf{P}'} = (f_1^\dagger, \cdots, f_{K^\dagger}^\dagger)$. Note that there is no guarantee for $K = K^\dagger$. So, we may choose a relatively large $K$, and hopefully a valid optimization algorithm will ultimately make $M_{k,1} = M'_{k,1} = 0$ (and hence $\widehat{SE}_{F_k|_1} = 0$) for some mimicked functions in $(F_1|_1, \cdots, F_K|_1)$. These mimicked functions capture no valid information, which will then be excluded from $\mathbf{f}^\dagger_{\mathbf{P}_\mathcal{D}, \mathbf{P}'} = (f_1^\dagger, \cdots, f_{K^\dagger}^\dagger)$.

Again, we have not yet came up with an optimization algorithm to maximize $\mathcal{O}_{\mathbf{f}_{\text{mimic}}}$ in formula 22. So, there is no experimental result in this paper. However, we do have some preliminary ideas: In each training step, we may consider $\mathcal{O}_{\mathbf{f}_{\text{mimic}}}$ as the reward. The agent is $(F_1, \cdots, F_K)$, and the environment comprises the sample sets $\{X_n\}_{n=1}^{N}$, $\{X'_n\}_{n=1}^{N}$, and $\mathcal{D}'_{\text{fix}} = \{X'_1, \cdots, X'_L\}$. Then, we may apply reinforcement learning algorithms to maximize $\mathcal{O}_{\mathbf{f}_{\text{mimic}}}$ (Kaelbling et al., 1996), which is our future research focus.

## C.2 Designed Machine Learning Pipeline

We believe that the most straightforward way to implement our approach is through a convolutional layer-wise pipeline on an image dataset, which is briefly exhibited in Figure 5.

Suppose we have a dataset containing M images $\{I_1, I_2, \cdots, I_M\}$, where each image is of the shape $H \times W \times C$ (i.e., height $\times$ width $\times$ channel number). In the first layer of our pipeline, suppose there is a convolutional filter cropping out $L \times L \times C$-shaped tensors with a stride of 1 (Riad et al., 2022). We regard each $L \times L \times C$ tensor as our sample vector $X$. This leads to a vector space $\mathbf{X}$

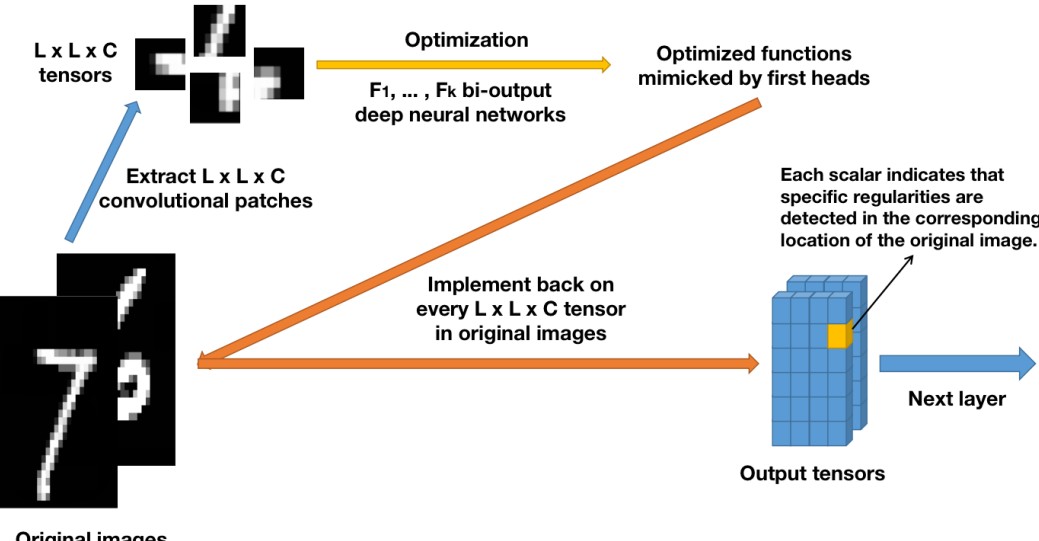

Figure 5: Convolutional layer-wise regularity learning pipeline: Convolutional patches will be extracted from original images, which are used as input vectors to optimize bi-output functions. Then, the optimized bi-output functions are implemented back onto the original images to generate output tensors from their first heads. These output tensors are used as inputs to next-layer optimization.

with dimension $C \cdot L^2$, and brings us a dataset $\mathcal{D}$ containing $M \cdot (H - L + 1) \cdot (W - L + 1)$ data samples. Without loss of generality, we assume that pixels in the original images are normalized to fall between 0 and 1, thereby defining our data space $\mathbf{S}$ to be the unit square within $\mathbf{X}$.

We assume that it is the data distribution $\mathbf{P}_\mathcal{D}$ that generates $\mathcal{D}$, and we obtain the uniform distribution $\mathbf{P}'$ on $\mathbf{S}$. Then, we initialize $K$ deep neural networks as the bi-output functions $F_1, \cdots, F_K$. After that, we apply the above approach to make $(F_1|_1, \cdots, F_K|_1)$, the sequence of mimicked functions by the first head of $F_1$ through $F_K$, converge to an optimal encoder with respect to $\mathbf{P}_\mathcal{D}$ and $\mathbf{P}'$. Suppose we can get the optimized sequence of mimicked functions $(F_1|_1^\dagger, \cdots, F_{K^\dagger}|_1^\dagger)$ with $K^\dagger \leq K$, where each mimicked function has observed spiking efficiency $\widehat{SE}_{F_k|_1^\dagger} > 0$.

Then, we apply $(F_1|_1^\dagger, \cdots, F_{K^\dagger}|_1^\dagger)$ to every $L \times L \times C$ convolutional patch in each original image $I_m$ among $\{I_1, I_2, \cdots, I_M\}$ again. That is, each mimicked function $F_k|_1^\dagger$ will be implemented on every $L \times L \times C$ tensor in an original image $I_m$ with stride 1. This brings us $M$ output tensors with shape $(H - L + 1) \times (W - L + 1) \times K^\dagger$, denoted as $\{O_1, O_2, \cdots, O_M\}$. Each dimension in an output tensor $O_m$ is the output scalar of one mimicked function in $(F_1|_1^\dagger, \cdots, F_{K^\dagger}|_1^\dagger)$. To be specific, each dimension in $O_m$ indicates that specific regularities are found in the corresponding area (i.e., the $L \times L \times C$ tensor) of the original image $I_m$.

Then, how does each dimension in $O_m$ distribute? Are there non-randomness and regularities in $\{O_1, O_2, \cdots, O_M\}$? Seeking for an answer, we may apply the same approach on $\{O_1, O_2, \cdots, O_M\}$: Each $\widetilde{L} \times \widetilde{L} \times K^\dagger$-shaped tensor in each $O_m$ is extracted in a convolutional manner. Then, we initialize $K$ new bi-output functions, whose first-head mimicked functions should be optimized to converge to an optimal encoder of the data distribution generating these $\widetilde{L} \times \widetilde{L} \times K^\dagger$ tensors. Implementing these optimized mimicked functions back on each $\widetilde{L} \times \widetilde{L} \times K^\dagger$ convolutional patch in each tensor $O_m$ among $\{O_1, O_2, \cdots, O_M\}$, we can further obtain the output tensors $\{\widetilde{O}_1, \widetilde{O}_2, \cdots, \widetilde{O}_M\}$, which is used as input for next level optimization.

We carry on this layer-wise optimization process until we are satisfied. Assuming the success of this process, the optimized mimicked functions in each level can explicitly learn regularities from different hierarchical levels of the image data. We believe that these learned optimal regularities can intuitively be regarded as vision (Cox & Dean, 2014; Kriegeskorte, 2015).

## D    THE REFINED THEORY TAKING SPIKING STRENGTH INTO CONSIDERATION

In this appendix section, we propose a more general theory, which we call the contour spiking theory, regarding how to learn regularities from data using spiking-level considered functions. Seeking for clarity, we refer to the theory in the main pages of this paper as simple spiking theory.

As we mentioned in Section 3.4, functions in our simple spiking theory cannot represent the data probability density variations within their spiking regions. Essentially, this is because we only consider spiking or not spiking of a function on an input vector, which is a relatively coarse strategy. Henceforth, we further consider the strength, or level, of spiking made by a function.

Suppose $\mathbf{X}$ is a finite-dimensional real or complex vector space, and suppose $\mathbf{S} \subset \mathbf{X}$ is a bounded sub-region within $\mathbf{X}$. We call a real scalar $\kappa > 0$ as the **grid**. Then, given a function $f : \mathbf{S} \to \mathbb{R}$, we say that $f$ makes a $l$-**level spiking** (or **spikes in the $l$ level**) on a vector $X \in \mathbf{S}$, if $l \cdot \kappa < f(X) \leq (l+1) \cdot \kappa$. When $l = 0$ (i.e., $0 < f(X) \leq \kappa$), we say that $f$ makes a **bottom level spiking** on $X$. Also, by choosing an integer $L \in \mathbb{N}$ to be the **top level**, we say that $f$ makes a **top level spiking** on vector $X$, if $f(X) > L \cdot \kappa$. Finally, we say that $f$ does not spike on $X$ if $f(X) \leq 0$, which can be regarded as a **-1 level spiking** (i.e., $l = -1$).

Then, suppose we have the data probability distribution $\mathbf{P}$ and random probability distribution $\mathbf{P}'$ defined on $\mathbf{S}$. With a large enough sampling size $N$, suppose we have $N$ data samples $\{X_n\}_{n=1}^{N}$ generated by $\mathbf{P}$ and $N$ random samples $\{X_n'\}_{n=1}^{N}$ generated by $\mathbf{P}'$. For $l = -1, 0, 1, \cdots, L$, suppose the function $f$ makes $M_l$ $l$-level spikings on the data samples $\{X_n\}_{n=1}^{N}$ and $M_l'$ $l$-level spikings on the random samples $\{X_n'\}_{n=1}^{N}$.

Accordingly, we can get the observed spiking probability distribution on random samples as $\widehat{P}' = (\frac{M_{-1}'}{N}, \frac{M_0'}{N}, \frac{M_1'}{N}, \cdots, \frac{M_L'}{N})$, which is also our null hypothesis on the data samples. But instead, we get the observed spiking probability distribution on data samples as $\widehat{P} = (\frac{M_{-1}}{N}, \frac{M_0}{N}, \frac{M_1}{N}, \cdots, \frac{M_L}{N})$. Similar to Section 3.1, the KL-divergence of $\widehat{P}$ over $\widehat{P}'$ is calculated by:

$$D_{KL}(\widehat{P}||\widehat{P}') = \sum_{l=-1}^{L} \frac{M_l}{N} \log(\frac{M_l}{M_l'}), \tag{23}$$

which measures the amount of information obtained if we replace $\widehat{P}'$ by $\widehat{P}$ to estimate the spiking probability distribution of $f$ on data samples Shannon (1948); Shlens (2014). Similar to Section 3.1, this is also the amount of information $f$ obtained from the data distribution $\mathbf{P}$ by comparing $\mathbf{P}$ with $\mathbf{P}'$. Accordingly, we define the **theoretical spiking efficiency** and **observed spiking efficiency** of $f$ to be:

$$SE_f = \lim_{N \to \infty} \left( \sum_{l=-1}^{L} \frac{M_l}{N} \log(\frac{M_l}{M_l'}) \right) \quad ; \quad \widehat{SE}_f = \sum_{l=-1}^{L} \frac{M_l}{N} \log(\frac{M_l + \alpha}{M_l' + \alpha}). \tag{24}$$

The **size** of function $f$, denoted as $|f|$, is still the number of adjustable parameters in $f$. The **conciseness** of $f$ is defined as $C_f = |f|^{-1}$. Then, the **theoretical ability** of $f$ is defined to be $A_f = SE_f \cdot C_f$, and the **observed ability** of $f$ is defined to be $\widehat{A}_f = \widehat{SE}_f \cdot C_f$:

$$A_f = \lim_{N \to \infty} \left( \sum_{l=-1}^{L} \frac{M_l}{N} \log(\frac{M_l}{M_l'}) \right) \cdot \frac{1}{|f|} \quad ; \quad \widehat{A}_f = \left( \sum_{l=-1}^{L} \frac{M_l}{N} \log(\frac{M_l + \alpha}{M_l' + \alpha}) \right) \cdot \frac{1}{|f|}. \tag{25}$$

The (theoretical and observed) ability of function $f$ is the ratio between the amount of information captured by $f$ from $\mathbf{P}$ and the amount of information possessed by the parameters in $f$. Also, the ability measures the effort made by $f$ to encode the captured information into its own parameters.

We define the $l$-**level spiking region** of $f$, denoted as $\mathbf{S}_{f,l}$, to be the set containing all vectors in $\mathbf{S}$ that make $f$ spike in the $l$-level. That is, $\mathbf{S}_{f,l} = \{X \in \mathbf{S} \mid l \cdot \kappa < f(X) \leq (l+1) \cdot \kappa\}$ for $l = 0, 1, \cdots, L-1$. The **top level spiking region** of $f$ is $\mathbf{S}_{f,L} = \{X \in \mathbf{S} \mid f(X) > L \cdot \kappa\}$. The **-1 level spiking region** of $f$, corresponding to $\mathbf{S} \backslash \mathbf{S}_f$ in the main pages, is $\mathbf{S}_{f,-1} = \{X \in \mathbf{S} \mid f(X) \leq 0\}$. It is easy to see that $\mathbf{S}_{f,l} \cap \mathbf{S}_{f,\tilde{l}} = \emptyset$ if $l \neq \tilde{l}$, and $\cup_{l=-1}^{L} \mathbf{S}_{f,l} = \mathbf{S}$. If we set the vector space $\mathbf{X} = \mathbb{R}^2$, one can

imagine that spiking regions in different levels of $f$ distribute like regions between contour lines on a geographic map. This is the reason for us to call the refined theory to be 'contour spiking theory'.

Then, we provide the refined version of Lemma 1, whose proof is very similar to Lemma 1 and not provided again:

**Lemma 1-refined.** *Suppose* $\mathbf{X}$ *is a finite-dimensional real or complex vector space, and suppose* $\mathbf{S} \subset \mathbf{X}$ *is a bounded sub-region in* $\mathbf{X}$. *Suppose* $f : \mathbf{S} \to \mathbb{R}$ *is a continuous function defined on* $\mathbf{S}$. *Then, for any chosen grid* $\kappa > 0$ *and top level* $L \in \mathbb{N}$, *the non-negative spiking regions* $\mathbf{S}_{f,0}, \mathbf{S}_{f,1}, \cdots, \mathbf{S}_{f,L}$ *of* $f$ *are always Lebesgue-measurable.*

Similarly, we provide the refined versions of Hypothesis 1 and Theorem 1:

**Hypothesis 1-refined.** *Suppose* $\mathbf{X}$ *is a finite-dimensional real or complex vector space, and suppose* $\mathbf{S} \subset \mathbf{X}$ *is a bounded sub-region in* $\mathbf{X}$. *Suppose* $f : \mathbf{S} \to \mathbb{R}$ *is a function defined on* $\mathbf{S}$ *with a finite size (i.e., there are finite adjustable parameters in* $f$). *Then, for any chosen grid* $\kappa > 0$ *and top level* $L \in \mathbb{N}$, *the non-negative spiking regions* $\mathbf{S}_{f,0}, \mathbf{S}_{f,1}, \cdots, \mathbf{S}_{f,L}$ *of* $f$ *are always Lebesgue-measurable.*

**Theorem 1-refined.** *Suppose* $\mathbf{X}$ *is a finite-dimensional real or complex vector space, and suppose* $\mathbf{S} \subset \mathbf{X}$ *is a bounded sub-region in* $\mathbf{X}$. *Suppose we have the data probability distribution* $\mathbf{P}$ *defined on* $\mathbf{S}$, *with the probability density function to be* $g(X)$. *Furthermore, suppose there exists an upper bound* $\Omega < \infty$, *such that* $g(X) \leq \Omega$ *for any* $X \in \mathbf{S}$. *Finally, suppose we have the random distribution* $\mathbf{P}'$ *to be the uniform distribution defined on* $\mathbf{S}$.

*Suppose we choose the grid* $\kappa > 0$ *and top level* $L \in \mathbb{N}$. *Then, under* $\kappa$ *and* $L$, *for any function* $f : \mathbf{S} \to \mathbb{R}$ *with a finite size* $|f|$, *its theoretical spiking efficiency* $SE_f$ *obtained with respect to* $\mathbf{P}$ *and* $\mathbf{P}'$ *is bounded by* $0 \leq SE_f \leq \Omega \cdot |\mathbf{S}| \cdot \log (\Omega \cdot |\mathbf{S}|)$, *where* $|\mathbf{S}|$ *is the Lebesgue measure of data space* $\mathbf{S}$.

One can proof the refined version of Theorem 1 using very similar methods as we described in Appendix A, through which we can get the formulas of $SE_f$ as (in which $g'(X) \equiv \frac{1}{|\mathbf{S}|}$ is the probability density function of the uniform distribution $\mathbf{P}'$, and $|\mathbf{S}_{f,l}|$ is the Lebesgue-measure of the $l$-level spiking region $\mathbf{S}_{f,l}$):

$$SE_f = \sum_{l=-1}^{L} \left( \int_{\mathbf{S}_{f,l}} g(X)\, dX \right) \log \left( \frac{\int_{\mathbf{S}_{f,l}} g(X)\, dX}{\int_{\mathbf{S}_{f,l}} g'(X)\, dX} \right) = \sum_{l=-1}^{L} \left( \int_{\mathbf{S}_{f,l}} g(X)\, dX \right) \log \left( \frac{|\mathbf{S}| \int_{\mathbf{S}_{f,l}} g(X)\, dX}{|\mathbf{S}_{f,l}|} \right).$$
(26)

Now, we describe the refined theory when applying multiple functions to the data distribution $\mathbf{P}$, where the random distribution $\mathbf{P}'$ is default to be the uniform distribution on the data space $\mathbf{S}$. Suppose we have a sequence of functions $\mathbf{f} = (f_1, \cdots, f_K)$, where each function $f_k : \mathbf{S} \to \mathbb{R}$ has a finite size $|f_k|$.

Given $N$ data samples $\{X_n\}_{n=1}^{N}$ generated by $\mathbf{P}$, suppose there are $M_{1,-1}, M_{1,0}, M_{1,1}, \cdots, M_{1,L}$ data samples that make function $f_1$ spike in the $-1, 0, 1, \cdots, L$ levels, respectively. Similarly, given $N$ random samples $\{X_n'\}_{n=1}^{N}$ generated by $\mathbf{P}'$, suppose there are $M_{1,-1}', M_{1,0}', M_{1,1}', \cdots, M_{1,L}'$ random samples that make function $f_1$ spike in the $-1, 0, 1, \cdots, L$ levels, respectively.

Then, suppose after ignoring all the data samples in $\{X_n\}_{n=1}^{N}$ making $f_1$ spike in any non-negative level, there are $M_{2,l}$ data samples in $\{X_n\}_{n=1}^{N}$ making $f_2$ spike in the $l$ level ($l \geq 0$). Similarly, suppose after ignoring all random samples making $f_1$ spike in any non-negative level, there are $M_{2,l}'$ random samples in $\{X_n'\}_{n=1}^{N}$ making $f_2$ spike in the $l$ level ($l \geq 0$). In general, we can get $M_{k,0}, M_{k,1}, \cdots, M_{k,L}$ data samples in $\{X_n\}_{n=1}^{N}$ and $M_{k,0}', M_{k,1}', \cdots, M_{k,L}'$ random samples in $\{X_n'\}_{n=1}^{N}$ that make $f_k$ spike in the $0, 1, \cdots, L$ levels respectively, but do not make $f_1, \cdots, f_{k-1}$ spike in any non-negative level. Then, we have $M_{k,-1} = N - \sum_{l=0}^{L} M_{k,l}$ as well as $M_{k,-1}' = N - \sum_{l=0}^{L} M_{k,l}'$.

In our opinion, spiking levels are our subjective classifications. Objectively, if the non-randomness is already discovered by functions $f_1, \cdots, f_{k-1}$ in the sequence $\mathbf{f}$, then such non-randomness cannot be awarded to $f_k$. So, we ignore any sample making any of $f_1, \cdots, f_{k-1}$ spike in any non-negative level, when counting the 'valid spikings' made by $f_k$.

Then, we can define the **theoretical spiking efficiency** $SE_{f_k}$ and **observed spiking efficiency** $\widehat{SE}_{f_k}$ for each function $f_k$ in $\mathbf{f} = (f_1, \cdots, f_K)$ as:

$$SE_{f_k} = \lim_{N \to \infty} \left( \sum_{l=-1}^{L} \frac{M_{k,l}}{N} \log(\frac{M_{k,l}}{M'_{k,l}}) \right) \quad ; \quad \widehat{SE}_{f_k} = \sum_{l=-1}^{L} \frac{M_{k,l}}{N} \log(\frac{M_{k,l} + \alpha}{M'_{k,l} + \alpha}) \qquad (27)$$

Then, suppose $M_{\mathbf{f},l} = \sum_{k=1}^{K} M_{k,l}$ and $M'_{\mathbf{f},l} = \sum_{k=1}^{K} M'_{k,l}$ for $l = 0, 1, \cdots, L$. We have that $M_{\mathbf{f},l}$ represents the number of data samples in $\{X_n\}_{n=1}^N$ making at least one function in $\mathbf{f} = (f_1, \cdots, f_K)$ spike in the $l$ level, while $M'_{\mathbf{f},l}$ represents the number of random samples in $\{X'_n\}_{n=1}^N$ making at least one function in $\mathbf{f}$ spike in the $l$ level. Also, suppose $M_{\mathbf{f},-1} = N - \sum_{l=0}^{L} M_{\mathbf{f},l}$ and $M'_{\mathbf{f},-1} = N - \sum_{l=0}^{L} M'_{\mathbf{f},l}$. The **theoretical spiking efficiency** and **observed spiking efficiency** of $\mathbf{f}$ is then defined by:

$$SE_{\mathbf{f}} = \lim_{N \to \infty} \left( \sum_{l=-1}^{L} \frac{M_{\mathbf{f},l}}{N} \log(\frac{M_{\mathbf{f},l}}{M'_{\mathbf{f},l}}) \right) \quad ; \quad \widehat{SE}_{\mathbf{f}} = \sum_{l=-1}^{L} \frac{M_{\mathbf{f},l}}{N} \log(\frac{M_{\mathbf{f},l} + \alpha}{M'_{\mathbf{f},l} + \alpha}) \qquad (28)$$

Accordingly, we can obtain the **theoretical ability** $A_{f_k} = SE_{f_k} \cdot C_{f_k}$ and the **observed ability** $\widehat{A}_{f_k} = \widehat{SE}_{f_k} \cdot C_{f_k}$ for each $f_k$ in the sequence of functions $\mathbf{f} = (f_1, \cdots, f_K)$, where $C_{f_k} = |f_k|^{-1}$ is the **conciseness** of $f_k$:

$$A_{f_k} = \lim_{N \to \infty} \left( \sum_{l=-1}^{L} \frac{M_{k,l}}{N} \log(\frac{M_{k,l}}{M'_{k,l}}) \right) \cdot \frac{1}{|f_k|} \; ; \; \widehat{A}_{f_k} = \left( \sum_{l=-1}^{L} \frac{M_{k,l}}{N} \log(\frac{M_{k,l} + \alpha}{M'_{k,l} + \alpha}) \right) \cdot \frac{1}{|f_k|} \qquad (29)$$

We define the **theoretical ability** of $\mathbf{f} = (f_1, \cdots, f_K)$ to be $A_{\mathbf{f}} = \sum_{k=1}^{K} A_{f_k}$. We define the **observed ability** of $\mathbf{f} = (f_1, \cdots, f_K)$ to be $\widehat{A}_{\mathbf{f}} = \sum_{k=1}^{K} \widehat{A}_{f_k}$:

$$A_{\mathbf{f}} = \sum_{k=1}^{K} \left[ \lim_{N \to \infty} \left( \sum_{l=-1}^{L} \frac{M_{k,l}}{N} \log(\frac{M_{k,l}}{M'_{k,l}}) \right) \cdot \frac{1}{|f_k|} \right]; \widehat{A}_{\mathbf{f}} = \sum_{k=1}^{K} \left[ \left( \sum_{l=-1}^{L} \frac{M_{k,l}}{N} \log(\frac{M_{k,l} + \alpha}{M'_{k,l} + \alpha}) \right) \cdot \frac{1}{|f_k|} \right]$$
$$(30)$$

Intuitively, $SE_{\mathbf{f}}$ measures the total amount of information captured by the sequence of functions $\mathbf{f} = (f_1, \cdots, f_K)$ from the data distribution $\mathbf{P}$, while $SE_{f_k}$ measures the amount of valid information captured by each $f_k$ in $\mathbf{f}$. Then, $A_{f_k}$ measures the effort made by each function $f_k$ to encode the valid information into the parameters, and $A_{\mathbf{f}}$ measures the total valid effort made by $\mathbf{f} = (f_1, \cdots, f_K)$ to encode the captured information into the parameters.

Then, the $l$-**level independent spiking region** of $f_k$ is defined to be $\mathbf{S}_{f_k,l}^{ind} = \{X \in \mathbf{S} \mid l \cdot \kappa < f_k(X) \le (l+1) \cdot \kappa\}$ for $l = 0, 1, \cdots, L-1$. The **top level independent spiking region** of $f_k$ is $\mathbf{S}_{f_k,L}^{ind} = \{X \in \mathbf{S} \mid f_k(X) > L \cdot \kappa\}$. The **-1 level independent spiking region** of $f_k$, corresponding to $\mathbf{S} \backslash \mathbf{S}_{f_k}$ in the main pages, is $\mathbf{S}_{f_k,-1}^{ind} = \{X \in \mathbf{S} \mid f_k(X) \le 0\}$.

Accordingly, for $l = 0, 1, \cdots, L$, the $l$-**level spiking region** of $f_k$ in the sequence of functions $\mathbf{f} = (f_1, \cdots, f_K)$ is defined to be $\mathbf{S}_{f_k,l} = \mathbf{S}_{f_k,l}^{ind} \backslash (\cup_{i=1}^{k-1} (\cup_{\tilde{l}=0}^{L} \mathbf{S}_{f_i,\tilde{l}}^{ind}))$ (i.e., removing all non-negative independent spiking regions of $f_1, \cdots, f_{k-1}$). That is, $\mathbf{S}_{f_k,l}$ consists of the vectors on which $f_k$ can make a valid spike in the $l$ level. It is easy to see that $\mathbf{S}_{f_k,l} \cap \mathbf{S}_{f_i,\tilde{l}} = \emptyset$ when $k \neq i$ **or** $l \geq 0 \neq \tilde{l} \geq 0$. Also, we have the **-1 level spiking region** of $f_k$ to be $\mathbf{S}_{f_k,-1} = \mathbf{S} \backslash (\cup_{l=0}^{L} \mathbf{S}_{f_k,l})$.

Finally, for $l = 0, 1, \cdots, L$, we define the $l$-**level spiking region** of the sequence of functions $\mathbf{f} = (f_1, \cdots, f_K)$ to be $\mathbf{S}_{\mathbf{f},l} = \cup_{k=1}^{K} \mathbf{S}_{f_k,l}$. That is, $\mathbf{S}_{\mathbf{f},l}$ consists of the vectors on which at least one function in the sequence $(f_1, \cdots, f_K)$ can make a valid spike in the $l$ level. Especially, we note that $\mathbf{S}_{\mathbf{f},l} \neq \cup_{k=1}^{K} \mathbf{S}_{f_k,l}^{ind}$, which is different from the simple spiking theory (see Section 3.3). Also, we have the **-1 level spiking region** of $\mathbf{f}$ to be $\mathbf{S}_{\mathbf{f},-1} = \mathbf{S} \backslash (\cup_{l=0}^{L} \mathbf{S}_{\mathbf{f},l})$. A straightforward demonstration is shown in Figure 6.

Regarding the bound of $SE_{\mathbf{f}}$ and each $SE_{f_k}$, we propose the refined version of Theorem 2:

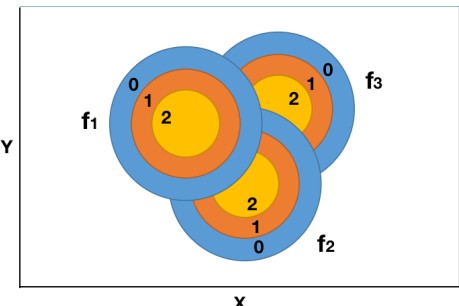

Figure 6: The leveled (contour) spiking regions of $\mathbf{f} = (f_1, f_2, f_3)$ defined on the xy-plane: In this example, we have $L = 2$, leading to 3 non-negative levels of spiking regions for each function. The 0-level spiking region for each $f_k$ (and also for $\mathbf{f}$) is in blue, the 1-level is in red, and the 2-level is in yellow. For each function $f_k$, its independent spiking regions in 3 non-negative levels form concentric circles. The non-negative level spiking regions of $f_1$ is in the front, while those of $f_2$ have to remove the regions overlapped with $f_1$, and those of $f_3$ have to remove the regions overlapped with both $f_1$ and $f_2$.

**Theorem 2-refined.** *Suppose $\mathbf{X}$ is a finite-dimensional real or complex vector space, and suppose $\mathbf{S} \subset \mathbf{X}$ is a bounded sub-region in $\mathbf{X}$. Suppose there are the data probability distribution $\mathbf{P}$ and the uniform distribution $\mathbf{P}'$ defined on $\mathbf{S}$. Also, suppose the probability density function $g(X)$ of $\mathbf{P}$ is bounded by a finite number $\Omega$ (i.e., $\mathbf{P}$ is regular).*

*Suppose $\mathbf{f} = (f_1, \cdots, f_K)$ is a sequence of functions with each function $f_k : \mathbf{S} \to \mathbb{R}$ possessing a finite size $|f_k|$. Suppose $\kappa$ is the chosen grid and $L$ is the chosen top level. Then, under $\kappa$ and $L$, with respect to $\mathbf{P}$ and $\mathbf{P}'$, the theoretical spiking efficiencies of both $\mathbf{f}$ and each $f_k$ are bounded. That is, we have $0 \le SE_{\mathbf{f}} \le \Omega \cdot |\mathbf{S}| \cdot \log(\Omega \cdot |\mathbf{S}|)$, as well as $0 \le SE_{f_k} \le \Omega \cdot |\mathbf{S}| \cdot \log(\Omega \cdot |\mathbf{S}|)$ for $k = 1, \cdots, K$. Here, $|\mathbf{S}|$ is the Lebesgue measure of data space $\mathbf{S}$.*

Similarly, we can follow the proof method of Theorem 1 to prove this refined version of Theorem 2. Also, we can get the formula of $SE_{\mathbf{f}}$ as (in which $g'(X) \equiv \frac{1}{|\mathbf{S}|}$ is the probability density function of the uniform distribution $\mathbf{P}'$, and $|\mathbf{S}_{\mathbf{f},l}|$ is the Lebesgue-measure of the $l$-level spiking region $\mathbf{S}_{\mathbf{f},l}$):

$$SE_{\mathbf{f}} = \sum_{l=-1}^{L} \left( \int_{\mathbf{S}_{\mathbf{f},l}} g(X) \, dX \right) \log\left( \frac{\int_{\mathbf{S}_{\mathbf{f},l}} g(X) \, dX}{\int_{\mathbf{S}_{\mathbf{f},l}} g'(X) \, dX} \right) = \sum_{l=-1}^{L} \left( \int_{\mathbf{S}_{\mathbf{f},l}} g(X) \, dX \right) \log\left( \frac{|\mathbf{S}| \int_{\mathbf{S}_{\mathbf{f},l}} g(X) \, dX}{|\mathbf{S}_{\mathbf{f},l}|} \right). \tag{31}$$

Replacing $\mathbf{f}$ by $f_k$ in the above formulas, we can then get the corresponding formulas of $SE_{f_k}$ for each function $f_k$ in $\mathbf{f}$. Then, similar to the simple spiking theory, two sequences of finite-sized functions $\mathbf{f} = (f_1, \cdots, f_K)$ and $\widetilde{\mathbf{f}} = (\widetilde{f}_1, \cdots, \widetilde{f}_{\widetilde{K}})$ will have the same theoretical spiking efficiency, if under the chosen grid $\kappa$ and top level $L$, their spiking regions $\mathbf{S}_{\mathbf{f},l}$ and $\mathbf{S}_{\widetilde{\mathbf{f}},l}$ in each non-negative level $l$ coincide with each other. Accordingly, we have:

**Definition 4-refined.** *Suppose $\mathbf{X}$ is a finite-dimensional real or complex vector space, and suppose $\mathbf{S} \subset \mathbf{X}$ is a bounded sub-region in $\mathbf{X}$. Suppose there are the data probability distribution $\mathbf{P}$ and the uniform distribution $\mathbf{P}'$ defined on $\mathbf{S}$. Also, suppose the probability density function $g(X)$ of $\mathbf{P}$ is bounded by a finite number $\Omega$ (i.e., $\mathbf{P}$ is regular). Finally, suppose $\kappa$ is the chosen grid and $L$ is the chosen top level for evaluating contour spikings.*

*Suppose $\mathbf{f} = (f_1, \cdots, f_K)$ and $\widetilde{\mathbf{f}} = (\widetilde{f}_1, \cdots, \widetilde{f}_{\widetilde{K}})$ are two sequences of finite-sized functions defined on $\mathbf{S}$. We say that under $\kappa$ and $L$, $\mathbf{f}$ and $\widetilde{\mathbf{f}}$ are **spiking equivalent** with respect to $\mathbf{P}$ and $\mathbf{P}'$, denoted as $\mathbf{f} \sim \widetilde{\mathbf{f}}$, if $SE_{\mathbf{f}} = SE_{\widetilde{\mathbf{f}}}$.*

*Suppose $\mathbf{f} = (f_1, \cdots, f_K)$ is a sequence of finite-sized functions defined on $\mathbf{S}$. Under $\kappa$ and $L$, we define the **spiking equivalence class** of $\mathbf{f}$, denoted as $\mathcal{E}_{\mathbf{f}}$, to be the set consisting of all the sequences of finite-sized functions that are spiking equivalent to $\mathbf{f}$. That is,*

$$\mathcal{E}_{\mathbf{f}} = \{\widetilde{\mathbf{f}} = (\widetilde{f}_1, \cdots, \widetilde{f}_{\widetilde{K}}) \big| |\widetilde{f}_k| < \infty \text{ for } k = 1, \cdots, \widetilde{K}; \text{ and } SE_{\mathbf{f}} = SE_{\widetilde{\mathbf{f}}}\},$$

*where $K$ and $\widetilde{K}$ are not necessarily equal, and different values of $\widetilde{K}$ are allowed in $\mathcal{E}_{\mathbf{f}}$.*

*Finally, under $\kappa$ and $L$, if there exists a sequence of finite-sized functions $\mathbf{f}^* = (f_1^*, \cdots, f_{K^*}^*)$, such that for any sequence of finite-sized functions $\widetilde{\mathbf{f}} = (\widetilde{f}_1, \cdots, \widetilde{f}_{\widetilde{K}})$, the inequality $SE_{\mathbf{f}^*} \geq SE_{\widetilde{\mathbf{f}}}$ always holds true, then we call $\mathbf{f}^*$ the **most efficient encoder** of $\mathbf{P}$ based on $\mathbf{P}'$, denoted as $\mathbf{f}_{\mathbf{P},\mathbf{P}'}^*$. We call $\mathcal{E}_{\mathbf{f}_{\mathbf{P},\mathbf{P}'}^*}$, the spiking equivalence class of $\mathbf{f}_{\mathbf{P},\mathbf{P}'}^*$, the **most efficient class** of $\mathbf{P}$ based on $\mathbf{P}'$, denoted as $\mathcal{E}_{\mathbf{P},\mathbf{P}'}^*$.*

Regarding the theoretical and observed abilities of $\mathbf{f} = (f_1, \cdots, f_K)$ as discussed earlier in this appendix section, here is the refined version of Hypothesis 4:

**Hypothesis 2-refined.** *Suppose $\mathbf{X}$ is a finite-dimensional real or complex vector space, and suppose $\mathbf{S} \subset \mathbf{X}$ is a bounded sub-region in $\mathbf{X}$. Suppose there are the data probability distribution $\mathbf{P}$ and the uniform distribution $\mathbf{P}'$ defined on $\mathbf{S}$. Also, suppose the probability density function $g(X)$ of $\mathbf{P}$ is bounded by a finite number $\Omega$ (i.e., $\mathbf{P}$ is regular). Finally, suppose $\kappa$ is the chosen grid and $L$ is the chosen top level for evaluating contour spikings.*

*Given a sequence of finite-sized functions $\mathbf{f} = (f_1, \cdots, f_K)$ defined on $\mathbf{S}$, suppose that under $\kappa$ and $L$, with respect to $\mathbf{P}$ and $\mathbf{P}'$, $\mathcal{E}_{\mathbf{f}}$ is the spiking equivalence class of $\mathbf{f}$, and $\Gamma = SE_{\mathbf{f}}$ is the theoretical spiking efficiency of $\mathbf{f}$. Then, there exists at least one sequence of finite-sized functions $\mathbf{f}^\dagger = (f_1^\dagger, \cdots, f_{K^\dagger}^\dagger) \in \mathcal{E}_{\mathbf{f}}$, such that for any other $\widetilde{\mathbf{f}} = (\widetilde{f}_1, \cdots, \widetilde{f}_{\widetilde{K}}) \in \mathcal{E}_{\mathbf{f}}$, the inequality $A_{\mathbf{f}^\dagger} \geq A_{\widetilde{\mathbf{f}}}$ always holds true. We call such a sequence of functions $\mathbf{f}^\dagger = (f_1^\dagger, \cdots, f_{K^\dagger}^\dagger)$ a $\Gamma$-**level optimal encoder** of $\mathbf{P}$ based on $\mathbf{P}'$, denoted as $\mathbf{f}_{\mathbf{P},\mathbf{P}'}^{\dagger \sim \Gamma}$.*

*Finally, suppose under $\kappa$ and $L$, the most efficient class $\mathcal{E}_{\mathbf{P},\mathbf{P}'}^*$ with respect to $\mathbf{P}$ and $\mathbf{P}'$ is not empty. Then, there exists at least one most efficient encoder $\mathbf{f}^\dagger = (f_1^\dagger, \cdots, f_{K^\dagger}^\dagger) \in \mathcal{E}_{\mathbf{P},\mathbf{P}'}^*$, such that for any other most efficient encoder $\mathbf{f}^* = (f_1^*, \cdots, f_{K^*}^*) \in \mathcal{E}_{\mathbf{P},\mathbf{P}'}^*$, the inequality $A_{\mathbf{f}^\dagger} \geq A_{\mathbf{f}^*}$ always holds true. We call such a sequence of functions $\mathbf{f}^\dagger = (f_1^\dagger, \cdots, f_{K^\dagger}^\dagger)$ an **optimal encoder** of $\mathbf{P}$ based on $\mathbf{P}'$, denoted as $\mathbf{f}_{\mathbf{P},\mathbf{P}'}^\dagger$.*

We can see that the refined versions of Theorem 1, Theorem 2, Definition 5 and Hypothesis 4 are almost the same as the original versions, except for adding the grid and top level regarding contour spiking. This indicates that the contour spiking theory and simple spiking theory are essentially the same. One can regard the simple spiking theory as a specific case of contour spiking theory when $L = 0$.

The refined version of Hypothesis 4 claims the existence of an optimal encoder under the chosen grid $\kappa$ and top level $L$. We note that the grid $\kappa$ and top level $L$ is chosen and then fixed in our discussion. There is another interesting question: Is it possible to find the optimal grid $\kappa^\dagger$ and optimal top level $L^\dagger$, so that an optimal encoder can provide the largest theoretical ability under $\kappa^\dagger$ and $L^\dagger$ among all other grids and top levels? This is within the scope of our future research.

# E    OPTIMAL ENCODERS BY EXAMPLES

In this appendix, we will provide different sequences of functions regarding the data distribution in each graph of Figure 2. We will show how the independent spiking regions regarding each sequence of functions divide the data space. Seeking for simplicity, in this appendix section, we always use the observed spiking efficiency with a large enough sampling size $N$ to approximate the theoretical spiking efficiency. That is, given the positive values $M$, $M'$ and $N$ as well as $\alpha = 10^{-10}$, we will calculate $\widehat{SE}(M, M', N)$ by:

$$\widehat{SE}(M, M', N) = \frac{M}{N} \log(\frac{M + \alpha}{M' + \alpha}) + \frac{N - M}{N} \log(\frac{N - M + \alpha}{N - M' + \alpha}), \tag{32}$$

which can be applied to a single function $f$, a sequence of functions $\mathbf{f} = (f_1, \cdots, f_K)$, or a function $f_k$ within the sequence $\mathbf{f}$ in order to approximate the theoretical spiking efficiency.

We start from the example data distribution in the left graph of Figure 2, which contains the data distribution $\mathbf{P}$ to be a uniform distribution within two disjoint circles: $\sqrt{(x - 2)^2 + (y - 2)^2} = 1$ and

$\sqrt{(x-5)^2 + (y-2)^2} = 1$. The data space $\mathbf{S}$ is the rectangle $\{x, y \mid 0 \le x \le 7, 0 \le y \le 4\} \subset \mathbb{R}^2$, and the random distribution $\mathbf{P}'$ is uniform on $\mathbf{S}$. Figure 7 shows the independent spiking regions of functions within six different sequences of functions being applied to this example. We note that in all the six cases, the spiking region of the entire sequence of functions always coincides with the two circles. By a routine derivation, we know this makes each sequence of functions the most efficient encoder with respect to $\mathbf{P}$ and $\mathbf{P}'$.

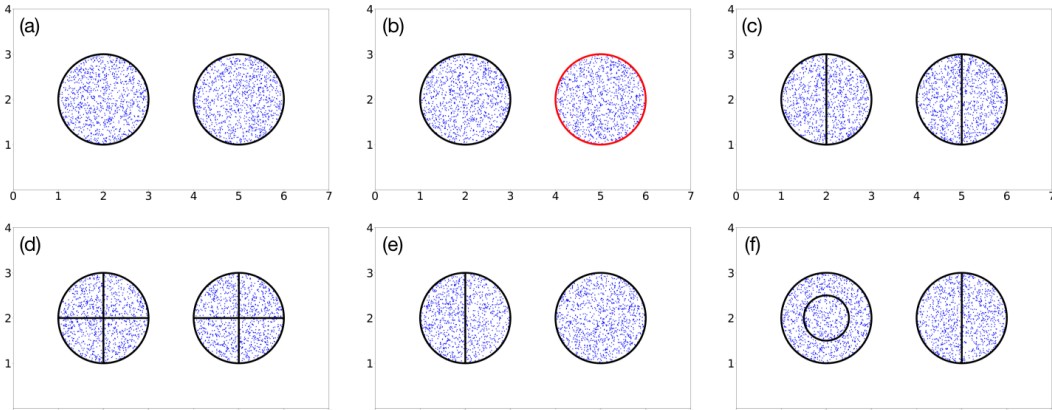

Figure 7: Independent spiking regions for different sequences of functions regarding the data distribution in the two circles.

• In graph (a) of Figure 7, there is only one function in the sequence. That is, $\mathbf{f} = (f_1)$, where

$$f_1(x, y) = \begin{cases} 1, & \text{if } \sqrt{(x-2)^2 + (y-2)^2} \le 1 \text{ or } \sqrt{(x-5)^2 + (y-2)^2} \le 1; \\ 0, & \text{otherwise.} \end{cases}$$

The area of each circle is $\pi r^2 = \pi$, whereas the area of $\mathbf{S}$ is $4 \cdot 7 = 28$. Suppose we choose the sampling size to be $N = 10000$. Then, there are approximately $M' = 10000 \cdot \frac{2\pi}{28} \approx 2244$ random samples generated by the uniform distribution $\mathbf{P}'$ that fall inside the two circles. On the other hand, all data samples must fall inside the spiking region of $f_1$, which is just the two circles. So, $M = 10000$. Finally, there are 6 adjustable parameters in $f_1$: the coordinates of each circle's center, namely, $(2, 2)$ and $(5, 2)$; as well as the radius, namely, 1 and 1. The exponent 2 is not adjustable, since adjusting its value will change the computational complexity of $f_1$. Hence, $|f_1| = 6$. As a result, we have the observed ability of $\mathbf{f} = (f_1)$ to be $\widehat{A}_{\mathbf{f}} = \widehat{SE}(10000, 2244, 10000)/6 \approx 0.249$.

• In graph (b) of Figure 7, there are two functions in the sequence: $\mathbf{f} = (f_1, f_2)$, where

$$f_1(x, y) = \begin{cases} 1, & \text{if } \sqrt{(x-2)^2 + (y-2)^2} \le 1; \\ 0, & \text{otherwise.} \end{cases} \quad , \quad f_2(x, y) = \begin{cases} 1, & \text{if } \sqrt{(x-5)^2 + (y-2)^2} \le 1; \\ 0, & \text{otherwise.} \end{cases}$$

Comparing to graph (a), only half data samples and random samples will fall inside one single circle. So, we have that for both $f_1$ and $f_2$, there are $M = 5000$, $M' \approx 1122$ and $N = 10000$. Also, we can get $|f_1| = |f_2| = 3$, since in each function there are three adjustable parameters: the circle's center $(x_c, y_c)$ and the radius $r$. Hence, we have $\widehat{A}_{\mathbf{f}} = \widehat{SE}(5000, 1122, 10000)/3 + \widehat{SE}(5000, 1122, 10000)/3 \approx 0.307$.

- In graph (c) of Figure 7, there are four functions in the sequence: $\mathbf{f} = (f_1, f_2, f_3, f_4)$, where $f_1$ through $f_4$ stands for each half circle from left to right:

$$f_1(x,y) = \begin{cases} 1, & \text{if } \sqrt{(x-2)^2 + (y-2)^2} \leq 1 \text{ and } x \leq 2; \\ 0, & \text{otherwise.} \end{cases}$$

$$f_2(x,y) = \begin{cases} 1, & \text{if } \sqrt{(x-2)^2 + (y-2)^2} \leq 1 \text{ and } x \geq 2; \\ 0, & \text{otherwise.} \end{cases}$$

$$f_3(x,y) = \begin{cases} 1, & \text{if } \sqrt{(x-5)^2 + (y-2)^2} \leq 1 \text{ and } x \leq 5; \\ 0, & \text{otherwise.} \end{cases}$$

$$f_4(x,y) = \begin{cases} 1, & \text{if } \sqrt{(x-5)^2 + (y-2)^2} \leq 1 \text{ and } x \geq 5; \\ 0, & \text{otherwise.} \end{cases}$$

Again, comparing to graph (b), half data samples and random samples will fall inside each half circle. So, we have that for $f_1$ through $f_4$, there are $M = 2500$, $M' \approx 561$ and $N = 10000$. Also, we can get $|f_k| = 4$ for $k = 1, \cdots, 4$, since in each function there are four adjustable parameters: the circle's center $(x_c, y_c)$, the radius $r$, and the diameter bar for the half circle. Hence, we have $\widehat{A}_{\mathbf{f}} = 4 \cdot \widehat{SE}(2500, 561, 10000)/4 \approx 0.201$.

- In graph (d) of Figure 7, we have $\mathbf{f} = (f_1, \cdots, f_8)$. Each $f_k$ has its spiking region to be one sector in the graph. We have $f_1$ through $f_4$ corresponding to the four sectors in the left circle in counter-clockwise. Similarly, $f_5$ through $f_8$ corresponds to those in the right circle in counter-clockwise. That is, $f_1(x,y) = 1$ when $\sqrt{(x-2)^2 + (y-2)^2} \leq 1$ and $x \leq 2, y \geq 2$; $f_2(x,y) = 1$ when $\sqrt{(x-2)^2 + (y-2)^2} \leq 1$ and $x \leq 2, y \leq 2$, etc. So, throughout $f_1$ to $f_8$, we have $M = 1250$, $M' \approx 280$ and $N = 10000$, whereas $|f_k| = 5$. Therefore, $\widehat{A}_{\mathbf{f}} = 8 \cdot \widehat{SE}(1250, 280, 10000)/5 \approx 0.152$.

- In graph (e) of Figure 7, there are two functions in the sequence $\mathbf{f} = (f_1, f_2)$, whereas their independent spiking regions are overlapped: $\mathbf{S}_{f_1}^{ind}$ is the right half of the left circle adding the whole right circle, while $\mathbf{S}_{f_2}^{ind}$ is the whole left circle. That is,

$$f_1(x,y) = \begin{cases} 1, & \text{if } \left( \sqrt{(x-2)^2 + (y-2)^2} \leq 1 \text{ and } x \geq 2 \right) \text{ or } \left( \sqrt{(x-5)^2 + (y-2)^2} \leq 1 \right); \\ 0, & \text{otherwise.} \end{cases}$$

$$f_2(x,y) = \begin{cases} 1, & \text{if } \sqrt{(x-2)^2 + (y-2)^2} \leq 1; \\ 0, & \text{otherwise.} \end{cases}$$

We have $M = 7500$ and $M' \approx 1683$ for $f_1$, since $\mathbf{S}_{f_1} = \mathbf{S}_{f_1}^{ind}$ covers 3 half circles. On contrast, the valid spikings made by $f_2$ are only within the left half of the left circle, corresponding to $\mathbf{S}_{f_2} = \mathbf{S}_{f_2}^{ind} \backslash \mathbf{S}_{f_1}^{ind}$. So, we have $M = 2500$ and $M' \approx 561$ for $f_2$. Also, $|f_1| = 3 \cdot 2 + 1 = 7$, since $f_1$ contains the center and radius of both circles as well as $x \geq 2$. Easy to see $|f_2| = 3$. Hence, $\widehat{A}_{\mathbf{f}} = \widehat{SE}(7500, 1683, 10000)/7 + \widehat{SE}(2500, 561, 10000)/3 \approx 0.184$.

- Finally, in graph (f) of Figure 7, there are four functions in the sequence $\mathbf{f} = (f_1, f_2, f_3, f_4)$: The independent spiking region $\mathbf{S}_{f_1}^{ind}$ covers the inner circle within the left circle, $\mathbf{S}_{f_2}^{ind}$ covers the whole left circle, $\mathbf{S}_{f_3}^{ind}$ covers the left half of the right circle, and $\mathbf{S}_{f_4}^{ind}$ covers the right half. That is:

$$f_1(x,y) = \begin{cases} 1, & \text{if } \sqrt{(x-2)^2 + (y-2)^2} \leq 0.5; \\ 0, & \text{otherwise.} \end{cases}$$

$$f_2(x,y) = \begin{cases} 1, & \text{if } \sqrt{(x-2)^2 + (y-2)^2} \leq 1; \\ 0, & \text{otherwise.} \end{cases}$$

$$f_3(x,y) = \begin{cases} 1, & \text{if } \sqrt{(x-5)^2 + (y-2)^2} \leq 1 \text{ and } x \leq 5; \\ 0, & \text{otherwise.} \end{cases}$$

$$f_4(x,y) = \begin{cases} 1, & \text{if } \sqrt{(x-5)^2 + (y-2)^2} \leq 1 \text{ and } x \geq 5; \\ 0, & \text{otherwise.} \end{cases}$$

By a routine analysis, we have $M = 1250$ and $M' \approx 280$ for $f_1$, $M = 3750$ and $M' \approx 842$ for $f_2$, as well as $M = 2500$ and $M' = 561$ for both $f_3$ and $f_4$. We have $|f_1| = |f_2| = 3$, while $|f_3| = |f_4| = 4$. Hence, $\widehat{A}_{\mathbf{f}} = \widehat{SE}(1250, 280, 10000)/3 + \widehat{SE}(3750, 842, 10000)/3 + 2 \cdot \widehat{SE}(2500, 561, 10000)/4 \approx 0.239$.

We can see that the largest observed ability $\widehat{A}_{\mathbf{f}}$ comes from graph (b). In this case, $\mathbf{f}$ contains two functions, and the independent spiking region of each function covers exactly one data distributed circle in the graph. By the above enumerating analysis, we can see that the sequence of functions in case (b) is likely an optimal encoder of $\mathbf{P}$ based on $\mathbf{P}'$, where the independent spiking regions of these two functions indeed divide the data space $\mathbf{S}$ in the most appropriate way with respect to the data distribution $\mathbf{P}$.

Then, we discuss the middle graph of Figure 2, in which the data distribution $\mathbf{P}$ is uniform within the area covered by two overlapped diamonds. The vertex of each diamond coincides with the center of the other diamond. The centers of the two diamonds are $x = 4, y = 3$ and $x = 6, y = 3$. Again, the random distribution $\mathbf{P}'$ is uniform within $\mathbf{S} = \{x, y \mid 0 \le x \le 10, 0 \le y \le 6\}$. In Figure 8, we show three sequences of functions regarding the data distribution in this example. Again, we use $\widehat{SE}(M, M', N)$ to approximate all encountered spiking efficiencies.

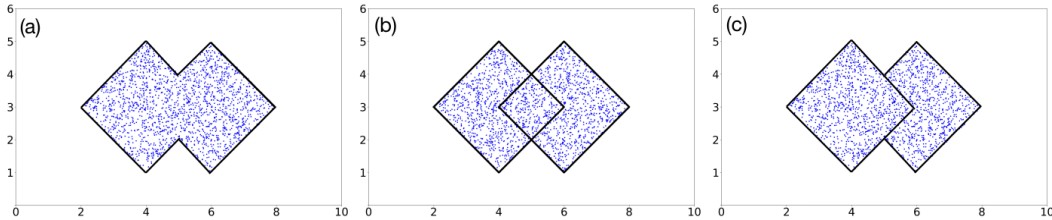

Figure 8: Independent spiking regions for different sequences of functions regarding the data distribution in the two overlapped diamonds.

• In graph (a) of Figure 8, the sequence only contains one function: $\mathbf{f} = (f_1)$. The spiking region of $f_1$ covers both diamonds. The boundary of $\mathbf{S}_{f_1}$ is indicated as in the graph. We have that:

$$f_1(x, y) = \begin{cases} 1, & \text{if } (x - 3 \le y \le x + 1 \text{ and } -x + 5 \le y \le -x + 9) \text{ or} \\ & (x - 5 \le y \le x - 1 \text{ and } -x + 7 \le y \le -x + 11); \\ 0, & \text{otherwise.} \end{cases}$$

Easy to see that the area of each diamond is $(2\sqrt{2})^2 = 8$. The overlapped area between two diamonds is 2, and their entire covered area is 14. The area of the data space $\mathbf{S}$ is $10 \cdot 6 = 60$. Suppose the sampling size is $N = 4200$, so there will be approximately $4200 \cdot \frac{14}{60} = 980$ random samples generated by $\mathbf{P}'$ that fall inside the two diamonds. On contrast, all data samples generated by $\mathbf{P}$ will fall inside the two diamonds. So, $M = 4200$, $M' = 980$ and $N = 4200$.

One may say that there are 16 adjustable parameters in $f_1$: Each $ax + b \le y \le cx + d$ contains 4 parameters, which in total contributes 16 ones. However, to stay consistent with the previous example regarding the two circles, we do not regard the scalars multiplied by $x$ as adjustable parameters (which are 1 and -1 in this example). In addition, we only need four parameters to accurately describe one diamond rotated from a square: the center $(x_c, y_c)$, the edge length $l$ and the rotation angle $\theta$. So, in our opinion, there are 8 parameters in $f_1$ describing the two diamonds, indicating $|f_1| = 8$. As a result, we have the observed ability $\widehat{A}_{\mathbf{f}} = \widehat{SE}(4200, 980, 4200)/8 \approx 0.182$.

• In graph (b) of Figure 8, we have two functions being contained in the sequence $\mathbf{f} = (f_1, f_2)$, where $f_1$ stands for the left diamond and $f_2$ stands for the right one:

$$f_1(x, y) = \begin{cases} 1, & \text{if } x - 3 \le y \le x + 1 \text{ and} \\ & -x + 5 \le y \le -x + 9; \\ 0, & \text{otherwise.} \end{cases} \quad , \quad f_2(x, y) = \begin{cases} 1, & \text{if } x - 5 \le y \le x - 1 \text{ and} \\ & -x + 7 \le y \le -x + 11; \\ 0, & \text{otherwise.} \end{cases}$$

We have $M = 2400$ and $M' = 560$ for $f_1$, since its spiking region covers the entire left diamond. Then, after removing the overlapped region, the spiking region of $f_2$ covers 3/4 of the right diamond.

So, we have $M = 1800$ and $M' = 420$ for $f_2$. Then, given $|f_1| = |f_2| = 4$, we have $\widehat{A}_{\mathbf{f}} = \widehat{SE}(2400, 560, 4200)/4 + \widehat{SE}(1800, 420, 4200)/4 \approx 0.223$.

● In graph (c) of Figure 8, the sequence still contains two functions: $\mathbf{f} = (f_1, f_2)$. But their independent spiking regions are no more overlapped: $f_1$ still stands for the left diamond, while $f_2$ removes the overlapped part from its independent spiking region. That is,

$$f_1(x, y) = \begin{cases} 1, & \text{if } x - 3 \le y \le x + 1 \text{ and} \\ & -x + 5 \le y \le -x + 9; \\ 0, & \text{otherwise.} \end{cases}, \quad f_2(x, y) = \begin{cases} 1, & \text{if } (x - 5 \le y \le x - 1 \text{ and} \\ & -x + 7 \le y \le -x + 11) \text{ and not} \\ & (x - 3 \le y \le x + 1 \text{ and} \\ & -x + 5 \le y \le -x + 9); \\ 0, & \text{otherwise.} \end{cases}$$

Again, we have $M = 2400$ and $M' = 560$ for $f_1$, as well as $M = 1800$ and $M' = 420$ for $f_2$. But this time we have $|f_1| = 4$ and $|f_2| = 8$. Hence, we have $\widehat{A}_{\mathbf{f}} = \widehat{SE}(2400, 560, 4200)/4 + \widehat{SE}(1800, 420, 4200)/8 \approx 0.178$.

We have that the largest observed ability comes from case (b), where each function in the sequence of functions stands for one diamond. This is also the most appropriate way, according to our intuition, for the independent spiking regions to divide the data space with respect to $\mathbf{P}$. One may argue that the spiking region in graph (a) is the most appropriate dividing of the data space. However, we claim that our mind will in fact automatically 'fill in' the missing edges within the two symmetrically overlapped diamonds, which coincides with the independent spiking regions of case (b). Anyway, the sequence of functions in case (b) is likely an optimal encoder with respect to $\mathbf{P}$ and $\mathbf{P}'$ in this example.

Finally, we discuss the right graph of Figure 2: There are 15 squares within $\mathbf{S} = \{x, y \mid 0 \le x \le 14, 0 \le y \le 8\}$, whereas each square has edge length 1. The data distribution $\mathbf{P}$ is uniform within these 15 squares, while the random distribution $\mathbf{P}'$ is uniform in $\mathbf{S}$. Then, Figure 9 shows two different sequences of functions: The left graph in Figure 9 shows a sequence containing only one function $\mathbf{f} = (f_1)$, where the independent spiking region of $f_1$ covers all 15 squares. The right graph in Figure 9 shows a sequence containing 15 functions $\mathbf{f} = (f_1, \cdots, f_{15})$, where the independent spiking region of each function covers exactly one square.

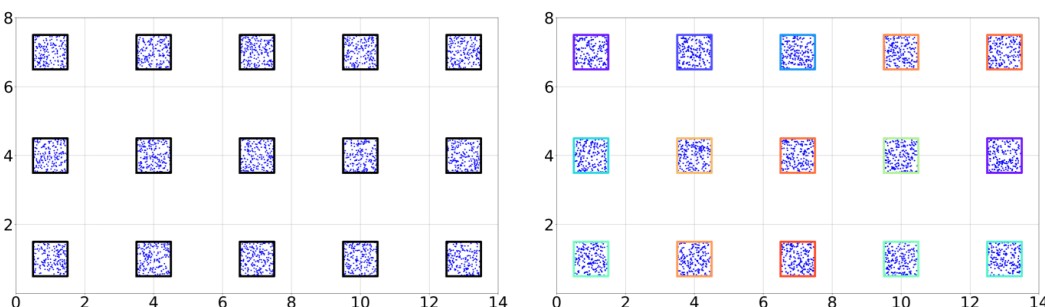

Figure 9: Independent spiking regions for two different sequences of functions regarding the data distribution in the 15 squares.

The area of $\mathbf{S}$ is $14 \cdot 8 = 112$, and the area of each square is 1. Suppose we choose $N = 11200$. Then, there are approximately $11200 \cdot \frac{15}{112} = 1500$ random samples falling inside all 15 squares, with each square containing around 100 random samples. So, in the left graph of Figure 9, we have $M = 11200$ and $M' = 1500$ for $f_1$. The size of $f_1$ is at least $3 \cdot 15 = 45$. This is because we need at least three parameters (the center $(x_c, y_c)$ and the edge length $l$) to accurately describe one square. Without lose of generality, we always work with optimal functions (see Appendix B) for all spiking regions. Hence, $|f_1| = 45$. As a result, the observed ability is $\widehat{A}_{\mathbf{f}} = \widehat{SE}(11200, 1500, 11200)/45 \approx 0.045$.

Then, for the right graph of Figure 9, each function $f_k$ in $\mathbf{f} = (f_1, \cdots, f_{15})$ has its independent spiking region covering exactly one square. We have that $M = 11200/15 \approx 747$ and $M' = 1500/15 = 100$ for each $f_k$. Also, $|f_k| = 3$ since each $f_k$ needs at least 3 parameters to record the center and edge length of each square. Hence, we have $\widehat{A}_{\mathbf{f}} = 15 \cdot \widehat{SE}(747, 100, 11200)/3 \approx 0.390$.

We also tried other sequences of functions, such as each function in the sequence corresponding to the squares in one row or one column. But none of them has an observed ability exceeding 0.39. Hence, the sequence of functions $\mathbf{f} = (f_1, \cdots, f_{15})$ corresponding to the right graph of Figure 9 is likely to be an optimal encoder with respect to $\mathbf{P}$ and $\mathbf{P}'$ in this example, in which the independent spiking regions of functions divide the data space in the most appropriate way with respect to $\mathbf{P}$.

As we mentioned in Section 3.4, in the last example, we want to discuss a data probability distribution that is not uniformly distributed within a region. As shown in Figure 10, the data distribution $\mathbf{P}$ has varied probability density within the two concentric circles: Its probability density within the inner circle $\sqrt{(x-4)^2 + (y-4)^2} \leq 1$ is 5 times higher than that within the outer circle $\sqrt{(x-4)^2 + (y-4)^2} \leq 2$. Again, $\mathbf{P}'$ is uniform on $\mathbf{S} = \{x, y \mid 0 \leq x \leq 8, 0 \leq y \leq 8\}$.

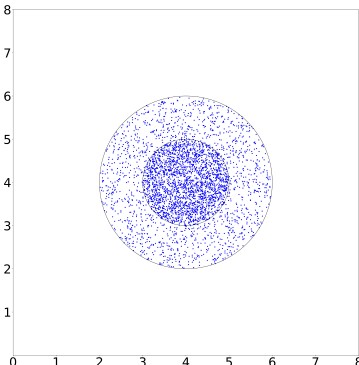

Figure 10: The data probability distribution whose probability density varies within two concentric circles.

Suppose we have the sampling size $N = 10000$. Then, there are approximately $10000 \cdot \frac{\pi}{64} \approx 491$ random samples falling in the inner circle, and approximately $10000 \cdot \frac{4\pi}{64} \approx 1963$ random samples falling in the outer circle. So, there are around 1472 random samples in the annular region. According to our setting, there are approximately $10000 \cdot \frac{5}{8} = 6250$ data samples falling in the inner circle, and approximately 3750 data samples in the annular region.

According to our enumeration, the most efficient encoder is likely $\mathbf{f} = (f_1)$, with the spiking region of $f_1$ covering the outer circle:

$$f_1(x, y) = \begin{cases} 1, & \text{if } \sqrt{(x-4)^2 + (y-4)^2} \leq 2; \\ 0, & \text{otherwise.} \end{cases}$$

We have the observed spiking efficiency of $\mathbf{f} = (f_1)$ to be $\widehat{SE}_{\mathbf{f}} = \widehat{SE}(10000, 1963, 10000) \approx 1.628$, which is the largest value we can get in our enumeration. Also, $\mathbf{f} = (f_1)$ is likely an optimal encoder in this example, whose observed ability is $\widehat{A}_{\mathbf{f}} = \widehat{SE}(10000, 1963, 10000)/3 \approx 0.543$. We have checked the observed ability of $\mathbf{f} = (f_1, f_2)$, where $f_1$ stands for the inner circle and $f_2$ stands for the outer circle. With the same theoretical (and observed) spiking efficiency as $\mathbf{f} = (f_1)$, this sequence of function $\mathbf{f} = (f_1, f_2)$ has a lower observed ability: $\widehat{A}_{\mathbf{f}} = \widehat{SE}(6250, 491, 10000)/3 + \widehat{SE}(3750, 1472, 10000)/3 \approx 0.466 < 0.543$. We failed to find another sequence of functions with an observed ability larger than 0.543 in this example. As a result, the obtained optimal encoder $\mathbf{f} = (f_1)$ ignores the inner circle, which fails to perfectly match with our intuition.

This example shows a defect of our theory: A spiking function cannot further represent the probability density variations within its spiking region. That says, we need to further consider how strong a function's spiking is, so that functions can encode the information from the data distribution in a more detailed way. This is just the contour spiking theory we proposed in Appendix D.

