# OpenReview forum: "Explainable self-supervised learning by spiking functions: a theory"
_ICLR.cc/2025/Conference — Submitted to ICLR 2025_

### Official Review · Reviewer_XGy7 · 2024-11-03

**Soundness:** 2
**Presentation:** 4
**Contribution:** 4
**Rating:** 5
**Confidence:** 3

**Summary:**

This paper proposes a novel theory to describe regularities in a data probability distribution.
1. Firstly, the authors define the spiking frequency of a function $f$ on a data set by calculating the ratio of points in the data set satisfying $f(x) > 0$.
2. Secondly, authors define the spiking efficiency of a function $f$ with respect to a data distribution $P$ and a random distribution $P'$ by the Kullback-Leibler divergence of the spiking distribution of $f$ on $P$ over that of $P'$.
3. Thirdly, the ability of $f$ is its spiking efficiency divided by its size.
4. Based on these concepts, the regularities in data are captured by the functions with the highest ability. Such functions can capture the largest amount of information from the data distribution and encode the captured information into the smallest amount of information.

**Strengths:**

1. The proposed theories aim to explain what are regularities. This is an important problem in self-supervised learning (and I believe it is also important in machine learning and mathematics) since it can help us understand what is learned by complicated models. To the best of my knowledge, existing works about explainable machine learning mainly focus on empirical experiments and visualization rather than rigorous theories.
2. The proposed theories are novel, explained in detail, and supported by sufficient examples.

**Weaknesses:**

It seems that Definition 2 is ambiguous. In Definition 2, authors define the size of a function $f$ as the number of adjustable parameters in $f$, where adjustable parameters are parameters that can be adjusted without changing the computational complexity of $f$.
1. It is hard (or at least not direct) to define which parameters can be adjusted without changing the computational complexity in mathematics. The first reason is that the current definition considers computational complexity, and the measurement of computational complexity is not clearly defined at least in the current version. The second reason is that a complicated function may have different expressions, and different expressions may lead to different computational complexities. Although we can define the computational complexity as the minimum one of all possible expressions, I think that such a definition is hard to calculate in practice since it is hard to traverse all possible expressions (which may be infinite). Based on these reasons, I think that Definition 2 is not rigorous.
2. There are many existing measurements of the complexity of function classes, such as the Vapnik–Chervonenkis dimension. However, the Vapnik–Chervonenkis dimension is defined for a function class, while Definition 2 considers the size of a single function. Is it possible to modify the proposed theory such that the Vapnik–Chervonenkis dimension can be used to measure the function complexity? The introduction of the Vapnik–Chervonenkis dimension may make the proposed theory more rigorous.

Authors consider Lebesgue measure throughout the paper but the presentation and proofs of theories can be more rigorous.
1. In Lemma 1, the set $S$ needs further conditions. The current version only requires bounded $S$, thus it does not exclude the case of immeasurable $S$. If $S$ is not Lebesgue measurable, then it is easy to construct a counterexample of Lemma 1.
2. The proof of Lemma 1 requires $S$ to be an open set, which is not mentioned in Lemma 1. In the proof, the authors claim that the ball $B(X,\delta)$, where $X$ satisfies $f(X) > 0$, is a subset of $S_f$. However, this only holds when $S$ is an open set, and $\delta$ is smaller than the distance between $X$ and the boundary of the open set $S$.

There is a minor mistake in line 313. In line 313, "for any function" should be "for some function" since $S_f$ consists of vectors that make at least one function in $f$ to spike, as explained in line 314.

**Questions:**

See weakness. My main concern is the first weakness.

---

> ### Author Response · Authors · 2024-11-26
> **Regarding non-measurable set S.**
>
> Thank you so much for your comments and reviews!
>
> Intuitively, the set S largely depends on human definition. That is, S is largely decided by the researchers. For instance, in MNIST dataset, if we divide all pixels by 255, we will get the data region S as the unit square on the 28*28 dimensional space (all dim ranges from 0 to 1). So, intuitively, there is no necessary to define a non-measurable set as S, which has no practical meaning but make our life harder. As a result, we always assume that S is measurable.
>
> Thanks again!

---

> > ### Comment · Reviewer_XGy7 · 2024-11-27
> >
> > Thanks for your response.
> >
> > Regarding the set $S$ (as discussed in the second paragraph of weakness), I persist that the description of Lemma 1 and its proof should be as rigorous as possible. As a theoretical paper, the theories depend on maths rather than human definitions. Although I admit that we mainly deal with ideal sets in practice, it is necessary to make the theory rigorous since it may be used in any possible case. Moreover, the authors mention non-measurable sets in the paper, thus it is natural to consider non-measurable cases in all theories.
> >
> > Regarding the size of a function (as discussed in the first paragraph of weakness), I am not convinced by the explanation provided in the response to reviewer ZevP. The proposed theory is not limited to the numerical estimation in this paper and can be used to explain any model. Moreover, the uniqueness of a definition is the foundation of discussing theories, while the definition of the size of a function (Definition 2) is still ambiguous after the authors' response.
> >
> > Based on the above reasons, I believe that the proposed theory needs further clarification (especially Definition 2). Thus, I change my score from 6 to 5.

---

### Official Review · Reviewer_Zz7H · 2024-11-03

**Soundness:** 3
**Presentation:** 2
**Contribution:** 2
**Rating:** 5
**Confidence:** 3

**Summary:**

The paper proposes a theoretical framework for explainable self-supervised learning through the use of spiking functions. The key objective is to explicitly capture regularities in data distributions to address the lack of interpretability in traditional deep learning. The authors present a new way to define and quantify non-randomness and regularities using information theory, and they propose optimization criteria for spiking functions to enhance explainability.

**Strengths:**

1. The introduction of spiking functions as a mechanism to formalize and quantify non-randomness in data is an interesting and novel contribution. The connection made between spiking neural networks (SNNs), information theory, and explainability could pave the way for a more formal approach to interpretable machine learning.
2. The paper develops a rigorous mathematical formalism for explaining the relationship between spiking functions, information capture, and conciseness. This formalism could potentially add value to how explainability is approached, particularly in self-supervised settings.

**Weaknesses:**

1. Though I appreciate a fully theoretical paper, this one stands out because the authors propose a new mechanism for explainable self-supervised learning through spiking functions—yet it lacks the essential empirical validation to support its claims. Without any concrete examples, simulations, or experimental evaluations, it is hard to assess the practical utility of the proposed framework, especially considering the complexity and abstraction of the presented mathematical formalism.

2. Theorems presented in the paper rely on several implicit assumptions, such as bounded data distributions and well-behaved spiking regions. These assumptions are not explicitly stated or discussed.

3. The mathematical formalism is highly abstract, with concepts like spiking efficiency and conciseness not being illustrated with practical examples or visualizations. This makes the ideas difficult to understand and limits accessibility for a broader audience.


In general, I like the idea of the paper but strongly believe it can be improved a lot more to meet its true potential.

**Questions:**

1. How would the authors implement these spiking functions in practice, and how would the spiking efficiency behave in an actual dataset? Including even simple experiments on synthetic data could significantly strengthen the validity of the claims.

2. There are several implicit assumptions that are critical to the proofs but are not explicitly discussed in the paper. For instance, Theorem 1 assumes that the probability density of the data distribution g(X) is uniformly bounded. In real-world datasets, distributions can have extreme values or unbounded regions. How do the authors envision handling situations where g(X) is not bounded or when the data density is highly irregular?

3. The paper introduces concepts like spiking efficiency, conciseness, and ability, but these terms remain abstract without concrete examples. It would be helpful to include examples or visualizations to illustrate how these quantities behave in a simple scenario. For instance, could the authors show what the spiking efficiency looks like in a 2D synthetic dataset?

4. Could the authors clarify the practical significance of these bounds? For example, do these bounds provide any guidance for selecting or designing spiking functions ? The implications of these results are not sufficiently discussed, and more clarity on this point would be beneficial.

---

> ### Author Response · Authors · 2024-11-26
>
> Thank you so much for your feedback and comments!
>
> Here are our answers to some of the questions:
> 1. In Appendix C, we described a pipeline to implement our theory on image datasets.
> 2. Thank you for pointing that out. In fact, we do not require the data distribution to be uniformly bounded. We just require no singularities in the data distribution. That is, we require no single points in the data region has infinite probability density. We believe that this requirement can be easily satisfied in practice: There is no image in MNIST, CIFAR or ImageNet possessing an infinite occurrence probability.
> 3. Some practical discussion and concrete examples are provided in Appendix C and E. We do not require the reviewers to read the appendices, but we do appreciate it very much if those sections can be evaluated.
> 4. Intuitively, the bound of data region largely depends on human definition. That is, the data region bound is largely decided by the researchers. For instance, in MNIST dataset, if we divide all pixels by 255, we get the data region S as the unit square on the 28*28 dimensional space (all dim ranges from 0 to 1). So, intuitively, researchers should define data region bound according to the features of the dataset. This is the significance of the bounds mentioned in our theory.
>
> Thanks again for your commends and reviews. We appreciate it so much!

---

### Official Review · Reviewer_ZevP · 2024-11-09

**Soundness:** 1
**Presentation:** 3
**Contribution:** 1
**Rating:** 1
**Confidence:** 5

**Summary:**

The paper describes a mathematical theory for explainable self-supervised learning through spiking functions. This theory seeks to make machine learning models more interpretable then end-to-end learning by defining and detecting non-randomness or regularities in data distributions  as compared to a base random distribution using spiking functions. Thereby also a new complexity measure for the functions is introduced and balanced against the spiking efficiency. The paper outlines how optimizing these functions can capture maximal information concisely, thus achieving an explainable, self-supervised learning system.

**Strengths:**

The definitions and derivation of the mathematical concepts is in itself correct. The mathematical arguments are clear and the formalisation is correct (I did not check the derivations of the appendix).

The author clearly show their understanding of the mathematics of the tools they use and the ability to formalize correctly. The paper is in itself written in a clear language style and explains the newly introduced concepts in an understandable way.

**Weaknesses:**

The paper discussed purely theoretical ideas, that reformulate ML theory, including complexity measures and non-randomness.
Bold statements need strong evidence to show its usefulness. In my opinion the authors failed to provide such evidence:
I cannot see the necessity for the connex with spiking networks. The f_k are simple indicator functions. The biological motivation is just anecdotal.

The presented non-randomness analysis is nothing else than classification learning, with P and P' being the sources of the two classes. SE is a reformulation of the empirical (in the limit the true) risk. The authors fail to show this IMHO trivial connection therefore overcomplicating the derivations.

There might be an important insight in the idea of an optimal encoder, yet due to the wired presentation as a new theory and due to the lack of any application the goes beyond the extremely simplified toy example, is is impossible for me to see the meaning and impact of that.

The presented research might be a ground start of a residual structural learning theory, if correctly set into the context of structural risk minimization, but it should be presented in a context (decision functions, VC theory, statistical learning theory) that makes it accessible.

**Questions:**

The analysis of the multiple function sequences could be acceptable as contribution to a residual learning framework. Yet the authors fail to clearly define corresponding loss functions for those residuals.

The idea of a "size" of a function, given by the number of its parameters is IMHO terribly flawed. The authors might consider VC-dimensions of hypothesis classes, instead of reinventing complexity measures here (problems with that kind of definition of a complexity measure: What about binary parameters, what about parameters that cancel out or are redundant like (a+b)*x, what about regularization effects). Maybe the idea of the "ability" of a function (and its optimization) should be put in relation to structural risk minimization (Vapnik).

I cannot see, why the authors use a KL divergence just for comparing the two ratios p_hat and p_hat' (which are just empirical risks). This artificial construction of a Bernoulli-distribution just seems to overcomplicate things.

If the individual elements of a sequence can be seen as "explanation" is IMHO largely dependent of what one wants to see as an explanation. There is no clear optimization goal defined, therefore it is impossible to judge if for real world examples those sequence elements would count as explanations. There is no discussion about the number K of functions in the sequence. The statement in line 482, that the distribution would determine the number only applies to the most trivial toy-examples and is by itself a complicated problem in the realm of optimal clustering (and clustering is precisely what the functions f_k are aiming for).

---

> ### Author Response · Authors · 2024-11-26
> **Regarding "size" of a function.**
>
> Thank you so much for your comments and feedback! We appreciate it so much.
>
> We know that the 'size' of a function can be defined in many different ways. But in our paper, we do not consider the symbolic complexities and other complexities. This is because under our definition (number of adjustable scalars), the numerical estimation in Section 3.4 works fine and coincides with commonsense. As a result, we simply keep the size of a function defined in such a way.
>
> We are currently working on feasible optimization algorithms to realize this theory, which depends on KL-divergence and other formulas. Thanks again for your comments on the optimization goal!

---

> > ### Comment · Reviewer_ZevP · 2024-12-02
> >
> > This debate about the „number of parameters“ as complexity measure alone is in my view already wired enough to clearly necessitate a  rejection. The recommended measure of VC-dimension captures precisely that notion and coincides with the authors „common sense“ for many simple cases. The hypothesis class emerges naturally from a parameterized function as the set of all discriminators that can be defined with any choice of the parameters from their respective definition sets.
> > The refusal of the authors to acknowledge this flaw in their approach shows the lack of connection to existing basic literature in machine learning theory (structural risk minimization).
> >
> > This authors failed to recognize and address the concern with the artificial use of KL divergence for the comparison of just two scalar values, which - according to the presented theory - will stay to be scalar values even in more complex scenarios.
> > KL-divergence is used to measure distances between distributions, which is by itself a complicated thing to do. For the comparison of two scalar values, simpler methods can be used and would be perfectly reasonable in this context. The refusal of the authors to acknowledge this flaw shows the lack of understanding and overview of the problem beyond the construction of (overcomplicated) formulas.
> >
> > The authors have not at all addressed the crucial concern about the goal of the required optimization and the choice of number K of residuals (or clusters).
> >
> > Overall this shows that the authors did not fall short in those respects in the paper just due to time constraints or oversight, but due to a lack of grounding in machine learning theory. I want to express that I state my opinion in such clarity in order to keep the authors from investing their time reinventing the wheel, when their work clearly shows that they would have the capacity and mathematical competence to be productive for the field, if they would ground their work in existing theory.
> >
> >
> > I will reduce my score and would strongly argue for the rejection of this paper.

---

### Official Review · Reviewer_VcgJ · 2024-11-09

**Soundness:** 3
**Presentation:** 3
**Contribution:** 2
**Rating:** 5
**Confidence:** 3

**Summary:**

The paper introduces a new mathematical theory that leverages spiking functions to encode non-random information as explainable regularities within self-supervised learning. The main aim is to establish a theoretical framework that uses spiking neural functions to differentiate data distributions from random noise. By applying information theory, the authors argue that optimal regularities — learned through the application of spiking functions — can support explainable self-supervised learning.

Approach: The authors construct a mathematical foundation to measure non-randomness by quantifying the information a spiking function can capture when applied to a data distribution. Key theoretical elements include spiking efficiency, conciseness, and ability of functions, with metrics based on the Kullback-Leibler divergence to quantify differences between data and random distributions. The theory is developed further by extending single spiking functions to sequences, allowing for layered information encoding. However, no empirical or implemented models are provided, with only theoretical demonstrations and hypothetical examples discussed.

Contribution: This work's theoretical contributions lie in redefining regularities and explainability in self-supervised learning using spiking functions. It could potentially provide the ICLR community with new perspectives on interpretability in self-supervised systems. However, the lack of empirical results or application contexts limits the practical insights offered by this submission.

**Strengths:**

- Novelty in Explainability: This paper proposes a unique approach to interpretability in self-supervised learning by defining explainable regularities through spiking functions.
- Mathematical Rigor: The theoretical basis is thorough, relying on information theory principles to quantify spiking efficiency and regularity, potentially valuable for theoretical advances.
- Significance: If validated empirically, this approach could influence future models of explainable self-supervised learning and contribute to theoretical advancements in spiking neural networks and information theory.

**Weaknesses:**

- Lack of Empirical Validation: The theory is purely speculative without an implemented model, making it difficult to assess practical effectiveness. Empirical results demonstrating its impact on existing self-supervised learning models would strengthen the paper.
- Implementation Feasibility: The authors acknowledge that practical implementation of their theory remains to be developed, leaving a significant gap between theory and application.
- Limited Practical Contribution: As the paper is theoretical, it lacks direct benchmarking that could contextualize it within existing explainability methods in machine learning.

**Questions:**

- In section 3,1, is there a typo in defining the theoretical spiking efficiency and observed spiking efficiency? The KL divergence definitions for both look the same. The difference between the two is understood after seeing equation 2 and 3.
- Could the authors discuss potential applications or contexts in which their theory might be validated empirically?
- What challenges do the authors foresee in implementing this framework for real-world self-supervised learning tasks?
- Has any thought been given to potential benchmarks or comparisons with existing self-supervised explainability models?

**Details Of Ethics Concerns:**

No ethics concerns.

---

> ### Author Response · Authors · 2024-11-25
> **Answer questions for**
>
> Thank you for your review. We appreciate it so much for the valuable feedback!
>
> Answer to Questions:
> 1. The definition of theoretical and observed spiking efficiency is correct. Both of them are defined based on the KL-divergence. But we add the small value alpha in the observed spiking efficiency to prevent dividing zero. Anyway, equation 2 and 3 is correct and as desired.
> 2. One potential implementation of our theory is image generation: We can apply our theory to refine autoencoders for image generation. The basic idea is to reconstruct the information in the image by as small amount of information as possible. Actually, the authors are currently working on this topic.
> 3. The biggest challenge should be a correct optimization method. That is, we need to find a way to update the network parameters, so that the deep network can not only capture information, but also represent the captured information by much smaller an amount of information. This optimization and information compression goal is challenging.
> 4. We have not yet came up with a good benchmark task for comparison. But this is definitely a great suggestion. Thank you so much!
>
> Thanks again for the comments and feedback.

---

### Official Review · Reviewer_JWyo · 2024-11-12

**Soundness:** 2
**Presentation:** 2
**Contribution:** 2
**Rating:** 3
**Confidence:** 3

**Summary:**

The paper presents a theory on learning regularities from data using spiking functions.

**Strengths:**

The theoretical framework presented in this paper is intriguing and can provide valuable insights, if backed up by empirical evidence.

**Weaknesses:**

The paper lacks empirical validation. The authors acknowledge that no implementation models or experimental results are provided. To strengthen their work, it will be beneficial  to develop and conduct experiments using benchmark datasets to demonstrate how the spiking functions perform in real-world applications.

The paper will also benefit from comparisons with existing explainable AI or self-supervised learning methods. It will be nice to position the approach of this paper within the context of current methodologies like contrastive learning models. Including a comparative analysis would help highlight the unique advantages and potential improvements offered by their spiking function-based theory.

The introduction of new metrics such as "theoretical spiking efficiency (SEf)" and "observed spiking efficiency (ŜEf)" would benefit from practical implementation guidance. The paper would benefit from detailed algorithmic outlines or pseudo-codes to illustrate how these metrics can be applied in real-world models.

**Questions:**

Please see weaknesses above.

---

### Meta-Review · Area_Chair_t1PZ · 2024-12-19

**Metareview:**

The authors claim to have a novel theory of interpretable machine learning built around spiking functions. The paper is entirely around a mathematical framework, with no experiments presented. The bar for a paper which consists entirely of a theoretical framework, without any new proofs or empirical results, is quite high. Especially if the authors claim that their framework says something useful about interpretability, then some results showing the learning of interpretable features is a must. The paper as is is not appropriate for a main conference, and should either be extended with experiments, or resubmitted to a more appropriate venue like a workshop or position paper track.

**Additional Comments On Reviewer Discussion:**

The reviewers all largely agreed that the lack of experimental results was an impediment to the paper, even if the framework sounded interesting. One reviewer went so far as to *lower* their score based on the discussion because they were concerned that the authors were not familiar enough with prior work on complexity measures to be able to justify the soundness of their new framework. While this was perhaps too harsh, I hope the authors will take the reviewer feedback into consideration.

---

### Decision · Program_Chairs · 2025-01-22

Reject